

# ERUO: a spectral processing routine for the MRR-PRO

Alfonso Ferrone[1], Anne-Claire Billault–Roux[1], and Alexis Berne[1]

[1]Environmental Remote Sensing Laboratory, École Polytechnique Fédérale de Lausanne (EPFL), Lausanne, Switzerland

**Correspondence:** Alexis Berne (alexis.berne@epfl.ch)

**Abstract.** The Micro Rain Radar (MRR) PRO is a K-band Doppler weather radar, using frequency modulated continuous wave (FMCW) signals, developed by Metek Meteorologische Messtechnik GmbH (Metek) as successor to the MRR-2. Benefiting from four datasets collected during two field campaigns in Antarctica and Switzerland, we developed a processing library for snowfall measurements, named ERUO (Enhancement and Reconstruction of the spectrUm for the MRR-PRO), with a two-fold

objective. Firstly, the proposed method addresses a series of issues plaguing the radar variables, which include interference lines, power drops at the extremes of the Doppler spectrum and abrupt cutoff of the transfer function. Secondly, the algorithm aims to improve the quality of the final variables, by lowering the minimum detectable equivalent attenuated reflectivity factor and extending the valid Doppler velocity range through antialiasing. The performance of the algorithm has been tested against the measurements of a co-located W-band Doppler radar. Information from a close-by X-Band Doppler dual-polarization radar

has been used to exclude unsuitable radar volumes from the comparison. Particular attention has been dedicated to verify the estimation of the meteorological signal in the spectra covered by interferences.

## 1   Introduction

While most of the densely populated areas of the planet benefit from the coverage by operational radar networks (Saltikoff et al., 2019), remote locations are often scarcely monitored. Among the latter, Antarctica constitutes an extreme example, with

precipitation often estimated from glaciological surface-based observations (Bromwich, 1990), numerical models (Bromwich et al., 2011), satellite products (Palerme et al., 2017) or measured exclusively at the ground by in-situ instruments (König-Langlo et al., 1998). This scarcity of measurements contrasts with the importance of precipitation and snowfall in particular, which constitute, together with water vapor deposition, the main water mass input to the ice sheet (Krinner et al., 2007). The same discourse, to a lesser extend, may be extended to the Arctic or to mountainous regions, where snowfall remains the

dominant precipitation type and measurements are hindered by technical and logistical difficulties.

In the last decades, however, field campaigns involving the deployment of small radars in such locations are becoming more common. One notable example is the Micro Rain Radar (MRR-2) (Klugmann et al., 1996), developed by Metek Meteorologische Messtechnik GmbH (Metek): a K-band (24 GHz), Doppler weather radar, using frequency modulated continuous wave (FMCW) signals. Multi-year datasets of its measurements are already available in several locations in Antarctica (Gorodet-

skaya et al., 2015; Grazioli et al., 2017), proving its suitability for deployment in hostile environments. Its potential for snowfall measurements has already been unlocked by the IMProToo algorithm (Maahn and Kollias, 2012), an alternative processing





technique for the raw MRR-2 spectra. By using an improved noise removal algorithm, their method is able to detect fainter meteorological signals, while the dealiasing allows for the recording of velocity values beyond the Nyquist velocity range, even though this eventuality is arguably rare in snowfall conditions. A different method, based on an alternative processing and dealiasing algorithm, has been proposed by Garcia-Benadi et al. (2020). Their technique also provides information on the dominant hydrometeor type in the observed radar values, and attempts to distinguish between precipitation types. A similar attempt, using two different classification algorithms, has later been proposed also by Foth et al. (2021).

The successor of the MRR-2 is the MRR-PRO: also developed by Metek, it is a K-band, Doppler, FMCW weather radar. Given its low power consumption, small size and relatively affordable cost, it is an ideal instrument for deployment in remote locations. Its usage for precipitation measurements has already been attested in the Antarctic peninsula (Pishniak et al., 2021) and on the Antarctic coast (Alexander, 2019). However, to our knowledge, no alternative processing procedures specific to this radar have yet been proposed to the scientific community. Theoretically, some of the processing algorithms designed for the MRR-2 may be re-adapted for the MRR-PRO. The first, arguably minor, problem that may be encountered in following this approach are the differences in file type and in the measurement configuration. In particular, some of the MRR-PRO configurable and fixed parameters differ from the ones of its predecessor, and any algorithms designed for the latter may need to be re-tuned.

The Environmental Remote Sensing Laboratory at EPFL had access to three of these radars, and in the last year they have been used for snowfall measurements in Switzerland and Antarctica. In all the datasets collected, we observed the presence of strong and semi-fixed signals, persisting for most of the duration of the campaign, at few radar-dependent range gates. These signals appear in all radar variables, and they are originating from spurious peaks in the raw spectra, spanning either a few or all the spectral line numbers. They appear even when the radar is deployed on the Antarctic plateau, hundreds of meters away from any tall orography and not surrounded by any structure taller than the MRR-PRO itself. To simplify the following discussion, we will thereafter refer to these anomalous signals as "interference lines". We suspect their origin to be a kind of interference internal to the radar, but we could not precisely identify it, and their technical investigation is beyond the scope of this study. These interference lines and some other issues with the radar data, described later throughout the manuscript, are the second (and main) reason why we decided not to re-adapt the MRR-2 algorithms to the MRR-PRO. These issues in the measurements need to be addressed by a new method, designed to remove them and specifically tuned on measurements from the MRR-PRO.

Therefore, this study presents a new processing algorithm for the MRR-PRO raw data, especially targeted at snowfall measurements, with a two-fold objective. Firstly, for a correct interpretation of the data, these interference lines need to be eliminated from all the variables. Thus, our method attempts to remove the spurious peaks from the raw spectra, reconstructing a clean spectrum and using it as starting point for the processing. Similarly, other issues such as the power drop observed at the extremes of the Doppler velocity range and the sporadic abrupt cut-off of the transfer function, used in the conversion from raw spectra to spectral reflectivity, are also addressed by the ERUO library. The second objective is an overall improvement of the quality of snowfall measurements: a different approach in the noise floor estimation ensures a better sensitivity, while the dealiasing allows for an extended range of detectable Doppler velocities. Snowfall has been chosen as the main target of our





library because its measurements can benefit significantly from improvements in sensitivity, and because the MRR-PRO files already contain a plethora of information specific to rain, thanks to the dedicated algorithm developed by Metek.

The method proposed in this study is named ERUO: Enhancement and Reconstruction of the spectrUm for the MRR-PRO.
This acronym has also been chosen for the meaning of the word "eruo" in classical Latin: depending on the context, it may literally be translated with the verb "to dig", and in figurative speech it may be interpreted as "to bring out". In a sense, our algorithm digs into the spectrum, extracting out the meteorological signal from below the interference lines covering it.

All the data and the additional instruments used in the analysis are presented in Section 2. The method is described in detail in Section 3, which is in turn divided in three subsections, each dedicated to one of the three main stages of the algorithm. The
validation of the proposed method is in Section 4, by comparing the output variables with the original MRR-PRO products and with the measurements of a second radar. A subsection has been dedicated specifically to the evaluation of the spectrum reconstruction, which is one of the innovative aspects of the proposed method. Finally, Section 5 contains our conclusions on the performances of the algorithm and its limitations.

## 2 Data

### 2.1 The MRR-PRO

The first three datasets used in this study have been collected by three MRR-PRO deployed in the vicinity of the research base Princess Elisabeth Antarctica (PEA) during the austral summer 2019-2020. The radars have been installed at three different locations, approximately between 7 and 17 kilometers apart from each other, across the mountains south of the base, in an effort to capture the evolution of precipitation systems at different stages of their interaction with the complex orography. To
better distinguish between the three instruments in the following discussion, we will refer to each system by its serial number: MRR-PRO 06, MRR-PRO 22 and MRR-PRO 23.

The fourth dataset is the result of a measurement campaign that took place in La Chaux-de-Fonds, in the Swiss Jura, during the boreal winter 2020-2021. The campaign was conducted in the framework of the Horizon2020 ICE GENESIS project (The ICE GENESIS Consortium, 2021), a collaboration of 36 partners from 10 countries, whose aim is to improve the representation
of snowfall in the perspective of aircraft safety. One of the radars used in PEA, the MRR-PRO 23, was deployed at the airport of the city. We will refer to this instrument as MRR-PRO ICE-GENESIS (shortened to ICEGENESIS in figures) when discussing matters pertaining to this campaign, to avoid confusion in the following text.

The MRR-PRO allows for a larger degree of customization in the measurement settings compared to its predecessor, the MRR-2. In both campaigns, all the MRR-PRO have been configured with the same exact parameters, listed in Table 1. Due
to constraints in the maximum allowed power consumption on site, the antenna heating had to be turned off for the whole deployment period.

For each dataset, the variables saved by the radars and used in the study are:





- $\mathbf{S}(t,i,n)$, the raw spectrum, dependent on the time ($t$), spectral line number ($i = 1,..,m$) and range gate number ($n = 1,..,n_{max}$) and measured in spectral units (S.U.), proportional to the return power;

- $\mathbf{Z_{ea}}(t,n)$, the attenuated equivalent reflectivity factor, expressed in dBZ;

- $\mathbf{V}(t,n)$, the Doppler radial velocity, in $\mathrm{ms}^{-1}$;

- $\mathbf{SW}(t,n)$, the spectral width, also in $\mathrm{ms}^{-1}$;

- $\mathbf{SNR}(t,n)$, the signal-to-noise ratio, in dB.

While variables collected by a specific radar will be written in bold roman font, as per convention when dealing with matrices,
the same symbol or acronym will be used in italic to denote the respective variable in general, without referring to a specific instance of it.

It should be noted that the MRR-PRO needs to be explicitly configured to save the raw spectrum rather than the reflectivity spectrum. ERUO is designed to function only with the former, even though we expect that only minor modifications would be needed to allow the support of the latter. In theory, it may be possible to convert the spectrum reflectivity back into raw
spectrum, and use the data directly as input for ERUO. However, this would require the knowledge of the exact procedure used to generate these two low-level variable, which we do not possess.

The variables listed above are also accompanied by a set of auxiliary quantities: $\mathbf{TF}(n)$, the transfer function, which expresses the receiver gate as a function of the range gate number, and $c$, the radar calibration constant used in the conversion from spectral density to spectral reflectivity. The usage of both variables is described in subsection 3.2.5.

**2.2   WProf**

During the ICE GENESIS campaign, a vertically-pointing W-band (94 GHz) Doppler cloud radar, thereafter referred to as WProf, was also deployed at the airport of La Chaux-de-Fonds, a few meters away from the MRR-PRO. This radar, officially known with the commercial name of RPG-FMCW-94-SP, is developed by Radiometer Physics GmbH (RPG). Its remarkable sensitivity, combined with the high resolution resulting from the usage of FMCW signals, makes it the perfect reference for
comparing the results of our algorithm later in this study.

This instrument allows a large degree of flexibility when deciding the configuration used for the data acquisition. In our case, we decided to use a chirp table divided in three stages, dividing the vertical extent in three consecutive sections, detailed in Table 2. Only a subset of the variables recorded by the instrument will be used in this study: $\mathbf{Z_{ea}^{W}}(t,n)$, $\mathbf{V^{W}}(t,n)$, and $\mathbf{SNR^{W}}(t,n)$. Each quantity is denoted by the same acronym as the MRR-PRO case, while the superscript $W$ is used to
distinguish the instrument that collected them.





### 2.3 MXPol

The last radar playing a role in this study is an X-band scanning Doppler dual-polarization weather radar (MXPol), developed by Prosensing. The instrument has been installed about 4.8 km away from the airport of La Chaux-de-Fonds during the ICE GENESIS campaign.

One of the scan types performed by this radar was a range height indicator (RHI) oriented towards the location of the MRR-PRO and WProf. Thanks to the dual-polarization nature of the measurements, we have access to an estimate of the hydrometeor types present in the observed volumes, computed using the classification described in Besic et al. (2016) and refined using the demixing algorithm described in Besic et al. (2018). The information on the hydrometeor population above the airport site is the only product of MXPol used in this study.

## 3   Method

This section presents the procedure followed by ERUO to process the raw spectra collected by the MRR-PRO. A schematic representation of it can be seen in Figure 1. As hinted by the scheme, the algorithm and its description are divided in three stages: preprocessing, processing and postprocessing, each described in a dedicated subsection below.

### 3.1   Preprocessing

The preprocessing uses the information contained in a whole dataset, collected from a single continuous deployment of the MRR-PRO, to compute three products:

- the interference line mask, $\mathbf{IM}(i,n)$, which highlights the regions of the raw spectra of the campaign likely affected by interference;

- the border correction, $\mathbf{BC}(i,n)$, a spectral density offset used to compensate the drop in the raw spectra values observed
at $i$ close to 1 or $m$;

- the raw signal-free spectrum profile estimate, $\mathbf{P}_e(n)$, a guess of the signal recorded in each range gate in ideal clear sky conditions, likely due to the thermal noise of the receiver.

     The procedure starts by loading all the raw spectra available in the dataset, which are concatenated along the temporal axis. From the resulting single 3-dimensional $(t, i, n)$ matrix, the algorithm extracts the 2-dimensional $(i, n)$ median raw spectrum
across all time steps, referred to as $\tilde{\mathbf{S}}(i,n)$. Panel a of Figure 2 shows how this median spectrum looks like for the MRR-PRO 06 dataset.

     In our case, the signature of precipitation was never visible in this matrix, being instead relegated to lower quantiles. However, it may be possible that other datasets contain a higher proportion of precipitation measurements, due to a higher frequency of events or to the lack of recording during prolonged clear-sky conditions. The user is advised to display $\tilde{\mathbf{S}}(i,n)$, and look for
the signature of precipitation in this matrix. If needed, a higher quantile can be used instead of the standard 0.5 value used to





compute $\tilde{\mathbf{S}}(i,n)$. This value can be easily changed by the user in the dedicated section of the configuration file of the ERUO library.

We were able to identify a set of characteristics recurrent across the median MRR-PRO spectra analyzed so far. First of all, the dependence of the median power from the range gate number seems to follow always the same pattern: after a sharp rise in the lowest gates, the power level briefly plateaus before starting a consistent descent that continues up to the last range gate. The exact position of the range gate in which the transition in trend happens may vary between MRR-PRO and campaigns. Another behavior consistently present in the $\tilde{\mathbf{S}}(i,n)$ matrix is the drop in raw spectrum observed at the extreme values of the spectral line number, already mentioned when introducing the quantity $\mathbf{BC}(i,n)$. This drop is, usually, of lower intensity when compared with other features, such as marked precipitation signal. Therefore it is extremely important to make sure that $\tilde{\mathbf{S}}(i,n)$ has been computed over a campaign with a relatively low frequency of precipitation (such as the ones used for this study) or over a subset of clear-sky measurements collected throughout the campaign. Finally, peaks of different size and entity, generating the interference lines in the final MRR-PRO products, may appear in several location across the median spectrum. Given the presence of these anomalies in both clear-sky and precipitation measurements, their signature is clearly visible in any quantile of $\tilde{\mathbf{S}}(i,n)$, including of course the default value of 0.5 used in this study. These peaks can be roughly divided in two categories:

– the isolated peaks, occupying only a handful of spectral lines and range gates, appearing as circular or oval shape when plotting $\tilde{\mathbf{S}}(i,n)$ versus both $n$ and $i$;

– the lines, covering the whole $i$ axis, from $i = 1$ to $i = m$, and spanning only a few range gates.

Two examples of the main category are visible in Figure 2, together with the other typical features described before. Additional interference lines of the second type are displayed in subsection 4.3, alongside a description of how the ERUO library handles their presence and a discussion on some possible impacts on the final radar variables.

Each of these observations plays a role in the preprocessing, allowing the use of the $\tilde{\mathbf{S}}(i,n)$ matrix as the basis from which the three products of this section will be derived. From this point onward, the algorithm continues following a two-step, quasi-iterative procedure. For clarity purposes, we will describe one iteration at a time.

### 3.1.1 First iteration

The main aim of this first step of the preprocessing is the elimination of isolated and anomalous spectral density peaks in $\tilde{\mathbf{S}}(i,n)$. Their presence can seriously impact $\mathbf{P}_e(n)$, by artificially raising the estimated clear-sky power level at the range gates affected by these spikes.

In order to identify these spurious signals, we first need to define a rough estimate of the background power level at each range gate. Therefore, a quantity akin to an early guess of $\mathbf{P}_e(n)$ is defined following this procedure:

1. The median of $\tilde{\mathbf{S}}(i,n)$ at each range gate, denoted by $\tilde{\mathbf{S}}(n)$, is computed. Its vertical gradient, $\nabla_n \tilde{\mathbf{S}}(n)$, is also estimated. Examples of both quantities for the MRR-PRO 06 dataset are shown in Figure 2.





2. Given the observed trend of the median spectrum, we decided to treat separately the lowest part of the profile (power increasing with height) and the second one (inverse trend). This step is necessary for the accurate identification of peaks in the spectrum, which is possible with our method only in the upper part of the profile, thanks to the small absolute value and limited fluctuations of its gradient. The first step is the identification of the range gate in which $\nabla_n \tilde{\mathbf{S}}(n)$ changes in sign, which provides a first guess of the beginning of the upper section of the profile. This estimate is further refined by excluding a few range gates at the beginning of the section, where $\nabla_n \tilde{\mathbf{S}}(n)$ is closer to zero and the raw spectrum plateaus. More precisely, the beginning of this second part of the profile is moved to the first $n$ in which $\nabla_n \tilde{\mathbf{S}}(n)$ reaches the median value of all negative $\nabla_n \tilde{\mathbf{S}}(n)$. We will refer to this range gate as $n_{up}$. Panel b of Figure 2 shows the location of $n_{up}$ for for the MRR-PRO 06, while panel c displays how the gradient does not reach its typical negative value immediately after the change of sign.

3. We now focus on the upper part of $\tilde{\mathbf{S}}(n)$, for $n > n_{up}$. This is the region most often affected by interference lines, which may lead to spurious peaks in the $\mathbf{P}_e(n)$ estimate. To exclude them, the algorithm imposes two thresholds on $\nabla_n \tilde{\mathbf{S}}(n)$. The maximum is set to 0, since the overall trend of the power profile in this region is towards lower values, and a positive derivative can only indicate the presence of a peak. The lower one instead aims to exclude the descending part of the spurious peak. The exact threshold value is set equal to the median $\nabla_n \tilde{\mathbf{S}}(n)$ for $n > n_{up}$ multiplied by a constant. This constant is set by default to 3, a value that gives satisfactory results for our datasets. As for all the parameters of ERUO, its value can be changed by the user in the configuration file.

4. We compute a smoother version of this upper part of the profile, by performing a polynomial fit of $\tilde{\mathbf{S}}(n)$, for $n > n_{up}$ and excluding the range gates that do not satisfy the thresholds described above. After some tests, the degree of the polynomial has been set to 4. While it appears that the fit is not very sensitive to small changes, such as increasing or lowering the degree by one, the chosen value seems to provide a curve that better maintains the profile trend, even if we were to extend the range to $n > n_{max}$. This may reduce the risk of contaminating the final part of the profile with the effect of a spurious inversion of the trend a few gates after the end of the fitted range. We will refer to the fit result as $\tilde{\mathbf{S}}_{fit}(n)$

5. Finally, we can define our early guess for the power received in each range gate in ideal clear sky conditions. Since the curve presented here is just a first estimate, we use the symbol $\mathbf{P}_e^{(1)}(n)$ for it. For $n \leq n_{up}$, we set $\mathbf{P}_e^{(1)}(n)$ equal to $\tilde{\mathbf{S}}(n)$. For the upper part, instead, we set $\mathbf{P}_e^{(1)}(n)$ equal to the element-wise minimum between $\tilde{\mathbf{S}}_{fit}(n)$ and $\tilde{\mathbf{S}}(n)$, to avoid the risk of artificially increasing the baseline power level.

A 2-dimensional matrix, $\tilde{\mathbf{S}}_{rec}^{(1)}(i,n)$, is created by tiling $m$ times the newly computed $\mathbf{P}_e^{(1)}(n)$ across a new dimension. This newly defined quantity represents a reconstruction of $\tilde{\mathbf{S}}(i,n)$ with a lower impact of the interference lines, since the strongest peaks are excluded from the input of the polynomial fit. This matrix, computed for the MRR-PRO 06, is shown in panel d of Figure 2. Even though it is only a preliminary estimate of the clear-sky raw spectrum, the reduced impact of interference and the lack of a drop at both ends of the Doppler velocity range can immediately be spotted.





Both features are enhanced when looking at the anomaly from this baseline raw spectrum, defined as $\mathbf{A}^{(1)}(i,n) = \tilde{\mathbf{S}}(i,n) - \tilde{\mathbf{S}}_{rec}^{(1)}(i,n)$. This matrix is used to compute $\mathbf{IM}^{(1)}(i,n)$, a partial interference mask, covering only the isolated anomalous spikes in the spectrum. Both $\mathbf{A}^{(1)}(i,n)$ and $\mathbf{IM}^{(1)}(i,n)$ (for MRR-PRO 06) can be seen in panel e of Figure 2.

The computation of $\mathbf{IM}^{(1)}(i,n)$ starts by masking (i.e. setting equal to 1) all regions where $\mathbf{A}^{(1)}(i,n) > 0.2$ S.U.. Once
again, this assumption is only valid if $\tilde{\mathbf{S}}(i,n)$ does not contain any leftover precipitation signal, either due to the relatively low frequency of precipitation, as in our case, or by being derived from a subset of clear-sky measurements collected throughout the campaign. In this way, we can be sure that the threshold does not affect regions of the spectrum containing precipitation signal for more than 50% of the campaign. Then, the columns $i = 1, 2, 3, m-2, m-1, m$ are all unmasked (i.e. set equal to 0), since the power drop at those spectral lines has not been corrected yet. Finally, all range gates containing at least $m-6$ masked
values are also artificially removed from the mask, since $\mathbf{IM}^{(1)}(i,n)$ aims to cover only the isolated peaks, and not the ones spanning the whole range $i = 1, ..., m$. Notice how in panel e of Figure 2 some of the interferences (especially the fainter ones) extend throughout the whole Doppler velocity range, while $\mathbf{IM}^{(1)}(i,n)$ only targets the central part of the spectrum.

The threshold 0.2 S.U. has been found to be the lowest one that allowed the masking of all the most visible interference in our four datasets, among all the tested values. The decision of excluding the three last spectral line numbers on the two
extremes of the velocity range is motivated by the drop in raw spectrum, more marked in those columns for all our datasets.

The border correction $\mathbf{BC}(i,n)$, the first real product of the preprocessing, is introduced to correct for this drop. After masking all regions of $\tilde{\mathbf{S}}(i,n)$ where $\mathbf{IM}^{(1)}(i,n) = 1$, the median of the remaining valid values at each range gate is computed and tiled $m$ times, resulting into a 2-dimensional matrix, $\tilde{\mathbf{S}}_{rec}^{(2)}$. This matrix is usually similar to the one displayed in panel d of Figure 2. However, while the latter is based on our reconstruction, $\tilde{\mathbf{S}}_{rec}^{(2)}$ is entirely based on a subset of the measured data.
Then, $\mathbf{BC}(i,n)$ is defined as equal to $\tilde{\mathbf{S}}_{rec}^{(2)} - \tilde{\mathbf{S}}(i,n)$ for the couples $(i,n)$ for which $\tilde{\mathbf{S}}_{rec}^{(2)} - \tilde{\mathbf{S}}(i,n) > 0$ S.U., and 0 everywhere else. Panel f of Figure 2 shows the result for the MRR-PRO 06.

### 3.1.2 Second iteration

The main product of the first iteration, $\mathbf{BC}(i,n)$, allows us to refine most of the previous estimates by reducing the impact of the power drop for $i$ close to 1 or $m$. Without this first estimate of the border correction, it would have been impossible to
accurately define an interference mask. The typical power drop at the extreme $i$ can be lower than 1 S.U., while the faintest of the interference lines can be constituted by anomalies as low as 0.2 S.U. , by definition. Therefore, the first can hide the second in some regions of the spectrum. This would result in the final products of the processing retaining these unmasked section of the interference lines, defying the whole purpose of the preprocessing.

As starting point of this second iteration of the preprocessing, we define a new quantity: $\tilde{\mathbf{S}}^{(3)}(i,n) = \tilde{\mathbf{S}}(i,n) + \mathbf{BC}(i,n)$.
Using it in place of $\tilde{\mathbf{S}}(i,n)$, we repeat the steps 1-5 of subsection 3.1.1. The resulting reconstructed profile is $\mathbf{P}_e(n)$, one of the final products of the preprocessing.

As in the previous subsection, a 2-dimensional matrix is derived by repeating $m$ times $\mathbf{P}_e(n)$ across a new dimension. This matrix is subtracted from $\tilde{\mathbf{S}}^{(3)}(i,n)$, giving us the new anomaly, $\mathbf{A}^{(2)}(i,n)$. To compute $\mathbf{IM}(i,n)$, the algorithm starts by masking (setting equal to 1) all entries $(i,n)$ where $\mathbf{A}^{(2)}(i,n) > 0.2$ S.U. If at a specific range gate number more than $0.9 \times m$





are masked, all the remaining values at that $n$ will be masked, too. Finally, the mask undergoes a series of 3 binary dilations. Each of them expands the mask to pixels adjacent to already masked ones (excludign the diagonal from the contiguity).

As mentioned when $\mathbf{IM}(i,n)$ was first introduced, the mask aims to cover the regions likely to contain interference. The latter appears in the spectrum as peaks or lines, whose exact position is changing during the campaign, and it may vary slightly in both $i$ and $n$. While the median spectrum, at the base of this preprocessing, captures the most likely position for the various

peaks and lines, variations in their positions may appear too seldom to leave a trace strong enough to be detected as interference. This is why the binary dilation and the threshold on the maximum number of masked spectral lines at any range gate have been introduced: they artificially expand the mask, resulting in a more likely coverage of these rare variations. The exact values for the parameters controlling the two operations have been set after a series of trials on our datasets. The final choice was a compromise: while the spectrum reconstruction (subsection 3.2.2) improves for smaller masked regions, the contamination by

interference in the final output of ERUO is reduced when the mask is expanded. The two parameters can also be modified by the user in the configuration file.

### 3.1.3 Loading large datasets

The size of the matrix resulting from the loading of all raw spectra is a consequence of both the frequency of acquisition and of the duration of the campaign. Depending on the hardware used at this stage, it may be impossible (or extremely tedious)

to load all the raw spectra at once, to compute $\tilde{\mathbf{S}}(i,n)$. Therefore, user discretion is advised at this stage, and some tests may be necessary to assess the limits of the computer used for the analysis. For reference, the laptop used for the development of ERUO had no difficulties in loading each of the four datasets presented in the Data section.

In case of problems we suggest two possible solutions:

- splitting the dataset in smaller chunks (one month duration at $10\,\mathrm{s}$ resolution should work on most commercially available
laptops), and run ERUO separately on each of them;

- manually selecting a sample of clear sky measurements, spaced throughout the campaign, and loading only those files at the beginning of the preprocessing.

### 3.2 Processing

This section describes how ERUO processes a single MRR-PRO file, using the raw spectra stored in it as starting point for

the computation of a set of radar variables: $\mathbf{Z}_{ea}^{(Proc)}(t,n)$, $\mathbf{V}^{(Proc)}(t,n)$, $\mathbf{SW}^{(Proc)}(t,n)$, and $\mathbf{SNR}^{(Proc)}(t,n)$. These quantities are stored in a new file, alongside the 3-dimensional matrix containing the spectral attenuated equivalent reflectivity, $\mathbf{SZ}_{ea}^{(Proc)}(t,i,n)$. As schematized in Figure 1, the processing can be seen as a series of short operations, each building on top of the output of the preceding one. These steps are detailed separately in the next subsections, following the same order as the library where the procedure is implemented. As usual, all numerical parameters introduced in the following subsections can be

customized by the user through the configuration file.





### 3.2.1  Transfer function reconstruction

The transfer function is used by FMCW radars to include the receiver gain dependence on the range gate number in the conversion of the raw spectra into spectral reflectivity. As visible in Figure 3, the codomain of these functions is approximately confined between 0 and 1.

In both datasets collected by the MRR-PRO 23 and used in this study, this function appears to have a sudden jump at $n = 128$ (approximately 2000 m above the radar in our configuration), with its value increasing by a factor of approximately $10^{38}$. The jump is displayed in Figure 3, represented by the almost vertical line connecting the last valid transfer function entry and the next, extremely high one. Therefore, the final radar variables available in the data file appear truncated at that range gate.

     Even though we cannot derive exactly what the value of the transfer function for $n > 128$ would have been without the 290   sudden increase, we decided to include in ERUO a step, preceding the real processing, that estimates it. By comparing the transfer function of the MRR-PRO 23 with the ones of the MRR-PRO 06 and MRR-PRO 22, we noticed that the location of the maximum for the former is at a significantly lower range gate than the ones of the other two MRR-PRO.

     We hypothesize that this shift and the lack of valid measurements above $n > 128$ are both a consequence of the incorrect handling, by the MRR-PRO 23, of the number of range gates set in its configuration. Since $n_{max}$ can be only increased or 295   decreased of a factor of 2, we suspect that the transfer function saved in the files is just a re-sampled version of the real transfer function over the wrong number of range gates, equal to the lower possible step in the radar configuration.

     Therefore, we decided to introduce an extra step in the processing routine, called transfer function reconstruction, to re-sample $\mathbf{TF}(n)$ over the correct $n_{max}$. This reconstruction is executed only if a problem analogous to the one observed in the MRR-PRO 23 dataset arises. Therefore, the algorithm starts by checking for entries in $\mathbf{TF}(n)$ above a certain threshold, set 300   by default to $9 \times 10^9$, and masks them. Then, using the "scipy.signal.resample" function (Virtanen et al., 2020), the unmasked section of the function is re-sampled over a number of samples equal to $n_{max}$. The result is finally re-scaled to ensure that the height of its maximum peak is the same as the unmasked section of the original function. In Figure 3, the final output for the MRR-PRO 23 (and ICEGENESIS) dataset is displayed as a brown line.

### 3.2.2  Spectrum reconstruction

From this point onward, the algorithm moves to the processing of each time step separately. The starting point of this spectrum-by-spectrum processing can be considered as the most delicate one of the whole ERUO library. As mentioned before, the removal of interference lines from the final products is among the topmost priority of the ERUO routine. Their manifestations in the raw spectrum, spurious peaks and lines, sometimes hide, in part or completely, the meteorological signal. During precipitation, they can artificially increase the amplitude of the recorded signal at the affected locations, resulting in the overestimation 310   of $Z_{ea}$ and, possibly, alterations of $V$ and $SW$, depending on the particular shape of the interference. Additionally, in clear-sky conditions, these peaks and lines generate anomalies that are recognized as signal by standard techniques such as the one proposed by Hildebrand and Sekhon (1974), and therefore often result in false positives of detected precipitation. Therefore, ERUO intervenes by identifying the regions of each spectra likely affected by interference, masks part of them and reconstructs



the masked regions using information from its surroundings. It should be noted that this part of the processing is optional, and

it can be avoided by setting the flag controlling it in the configuration file equal to 0. However, we strongly recommend running the spectrum reconstruction, to keep the output products as clean as possible.

For each time step $t_j$ of the input file, the process starts by loading of the raw spectra and summing $\mathbf{BC}(i,n)$ to it, obtaining the corrected raw spectra, $\mathbf{S}(t_j,i,n)$. An ideal clear-sky guess of the power return is defined, similarly to the preprocessing case, by repeating $m$ times along a new dimension the profile $\mathbf{P}_e(n)$, obtaining $\mathbf{S}^{(cs)}(t_j,i,n)$. This matrix is used to compute

the power anomaly, defined as $\mathbf{A}(t_j,i,n) = \mathbf{S}(t_j,i,n) - \mathbf{S}^{(cs)}(t_j,i,n)$ The first guess of the regions of this matrix affected by interference is obtained by selecting the couples $(i,n)$ where $\mathbf{IM}(i,n) = 1$ and $\mathbf{A}(t_j,i,n) > 1.0$ S.U.. The chosen threshold is significantly larger than the one used as minimum prominence for the signal detection, to avoid creating vast regions of masked values. As discussed later in the result section, the reconstruction performs better on smaller masked areas. The main aim of this reconstruction is the removal of the most noticeable interferences, which are usually associated to peaks taller than

the minimum detectable prominence threshold. Therefore, this choice of threshold suits the objective, while smaller, leftover interference is easily eliminated by the postprocessing.

This first set of locations can be seen as a collection of few contiguous regions of masked $(i,n)$ couples. Among those regions, only the ones that satisfy one of two conditions are kept:

- the masked section is an isolated peak. In practice, $\mathbf{A}(i,n) > 1.0$ s.u. for less than 5 couples $(i,n)$ in the surroundings

of the current masked region. The surrounding area in this case is computed by dilating twice the current contiguous masked region.

- for some of the affected $n$, the masked region covers at least 80% of the interval $i = 1,..,m$. This may indicate the presence of an interference spanning all spectral line numbers, extreme folding of the meteorological signal or particularly wide spectra, such as the ones recorded in presence of strong turbulence.

The second set of masked regions is further refined by unmasking, at each range gate number, the entry $(i,n)$ with the largest value of $\mathbf{A}(i,n)$, if strong anomalies are detected immediately above or below the region. This condition allows the preservation of the position and intensity of the maximum return power at each rage gate, constraining the following reconstruction. In practical terms, the algorithm first checks if $\mathbf{A}(i,n) > 5.0$ S.U. for at least one entry $(i,n)$ within the masked region. The second condition is the existence of locations $(i,n)$ satisfying the same condition in (at least) 3 of the 5 closest range gates

below or above the masked region. If the median $i$ coordinate of these peaks differs from the $i$ coordinate of the maximum peak of $\mathbf{A}(i,n)$ within the masked region by 5 or less, the algorithm proceeds with the unmasking of the location of the largest anomaly at each range gate.

The matrix $\mathbf{A}(i,n)$ is copied and, in this copy, all the previously masked region are substituted by the value "Not a Number" (NaN). These missing region are reconstructed using the "astropy.convolution.interpolate_replace_nans" function of the

Python3 library Astropy (Astropy Collaboration et al., 2013, 2018). As described in the documentation, this function replaces the NaN values with a kernel-weighted interpolation from their neighbors. We decided to use a Gaussian kernel for the reconstruction, after performing a few tests with the different types available in the library. While along the $i$-axis the kernel standard





deviation has been fixed to 1, on the $n$-axis the value is defined by the number of consecutive range gates containing at least one NaN divided by a scaling factor, set by default to 3. The kernel size is then left to the Astropy default value, which is eight times the standard deviation. Therefore, on the $n$-axis, the kernel always contains at least a not-NaN value. On the $i$-axis, a visual inspection revealed that a standard deviation larger than 1 often causes an artificial broadening of the reconstructed peaks, when compared to the precipitation signal directly above and below. Finally, the first 15 range gates are excluded from the whole procedure, due to the difficulties encountered in accurately reconstructing the peculiar behavior of the $\mathbf{A}(i,n)$ in the lowest gates, especially at the extremes of the $i$-axis.

### 3.2.3 Peak detection and dealiasing

The reconstructed anomaly is added back to the baseline clear-sky return, $\mathbf{S}^{(cs)}(t_j, i, n)$, and it is converted to linear units. An example of such product can be seen in the panel a of Figure 4. The raw spectrum in linear units is repeated three times along the $i$ axis, whose limits now become $[-m, 2 \times m]$. The resulting matrix is denoted by $\mathbf{H}(t_j, i, n)$, and its units by s.u.. Panel b of Figure 4 exemplifies the enlarged matrix. The repetition of the spectrum aids the future dealiasing, similarly to the technique proposed for the MRR-2 in Garcia-Benadi et al. (2020).

At each range gate number, the algorithm identifies all local maxima in the spectrum. If the largest maxima (usually three) have a prominence of at least 0.2 s.u., they are saved in the list of possible signal peaks. Note that features are usually found in sets of three, given the three repetition of the original spectrum along the $m$-axis. Secondary maxima are also saved, but only if their prominence is at least 25% of the main ones. While the absolute threshold is kept purposefully low, to detect even the faintest traces of precipitation signal, the relative one avoids the proliferation of false positives. Panel c of Figure 4 provides an example of peak detection for a spectrum collected by the MRR-PRO 06. The peaks with the largest prominence are highlighted in green, while the ones eliminated by the absolute and relative prominence threshold are displayed in red and blue, respectively. In the example, the blue peaks would have been (barely) acceptable, if no larger one was detected in the spectrum. This allows signals as low as the blue ones of panel c of Figure 4 to be detected if no clearer peak is present, while at the same time preventing them from interfering with the next step when stronger signal exists.

Once all the $(i, n)$ coordinates of the peaks are identified, the algorithm proceeds by trying to connect them in vertical lines. For each candidate peak, the algorithm looks in a window of 5 along the $n$-axis and 10 on the $i$-axis around its coordinate. If another peak falls in this window, the two are connected in a line, and the procedure continues by trying to connect more peaks to the same line. Each peak can belong only to one line, each line can contain only a single peak per range gate, and multiple lines can occupy the same $n$ (for example, in case of bimodality in the spectrum). If multiple peaks fall in the same window, the choice always favors the one in the closest range gate, using the distance along the $i$-axis as secondary decision parameter. Following with the example of Figure 4, four lines have been detected in this specific case, and they have been superimposed to $\mathbf{H}(t_j, i, n)$ in panel b.

Once the lines have been determined, the ones with less than 3 elements are excluded from the analysis. Among the remaining ones, the algorithm tries to exclude the duplicate ones caused by the three repetitions of the original spectrum. Couples of lines whose median difference along the $i$-axis is close to $m$ (the tolerance is $1\ \mathrm{ms^{-1}}$, converted in spectral line numbers) are flagged





as duplicates. Among these couples, the algorithm keeps the line whose upper half is closer, in median, to $i = 0$. This operation per-se does not count as full dealiasing. However, given the focus of ERUO on snow and the usual slow fall-speed of solid hydrometeors, especially in the upper part of the precipitation systems, this choice results in a reasonably good handling of
many of the aliasing cases of our four datasets.

Once all duplicates have been removed, the line that spans the largest number of range gates is considered the main one, and all the ones at a distance larger than $m$ along the $i$-axis are eliminated. Figure 4, in panel d, displays the resulting selection and the main line for the example spectra.

### 3.2.4 Signal identification

The set of lines derived in the previous section denotes only the position of the main peak in the spectrum at each range gate. To proceed with the analysis, it is necessary to extract $m$ spectral lines around each of these peaks. The selection of the section of the spectrum to keep at each range gate starts from the peak, expanding symmetrically from it in decreasing order of power.

Once the spectrum at each $n$ has been determined, the signal is separated from the noise following the Decreasing Average (DA) method. This technique, already used by Maahn and Kollias (2012) for the processing of some of the MRR 2 measure-
ments, appears more reliable than Hildebrand and Sekhon (1974) in our case. In short, the method starts by flagging the highest peak in the spectrum as signal. In a series of iteration, adjacent peaks are flagged, too, favoring always the largest one, until the average power of the remaining, unflagged spectrum continues to decrease. In our case, we stop the algorithm slightly before, by setting a minimum threshold of 0.001 s.u. for the decrease between consecutive iterations. This small modification prevents a problem that we often encountered with the original version, where the algorithm included sometimes the whole spectrum in
the signal.

Once the borders of the signal have been determined, we compute the noise level and its standard deviation (along the $i$ dimension). In the unfortunate case in which the signal has been found to occupy the whole spectrum, such as in the case of extreme folding or some leftover interference line, the noise level and standard deviation are set equal respectively to the minimum of the signal and 0 s.u.. The resulting profile of noise level, $\mathbf{NL}^{(1)}(t_j, n)$, undergoes further refinement. One of the
products of the preprocessing, $\mathbf{P}_e(n)$, gives us the median raw spectrum per range gate for the campaign in logarithmic form. After we convert it to linear units, we can compare it to $\mathbf{NL}^{(1)}(t_j, n)$, flagging all $n$ at which the computed noise level is more than 0.2 s.u above $\mathbf{P}_e(n)$. A copy of $\mathbf{NL}^{(1)}(t_j, n)$ with the entries at those range gates substituted by NaN is created, and convoluted with a box kernel 5 gates wide. The aim of this operation is the removal of gates potentially contaminated by remaining interference. The convolution is performed using, once again, the Astropy library, which has been set to perform a
prior interpolation in the NaN regions. The resulting smoothed noise level is merged with the original $\mathbf{NL}^{(1)}(t_j, n)$ only in the flagged range gates, giving us the final noise level, $\mathbf{NL}(t_j, n)$.

### 3.2.5 Moments computation

The noise level and standard deviation identified in the previous section allow us to proceed with the computation of the final product of the processing section.





The process starts by adding the noise standard deviation multiplied by 3 to the noise level, as a mean to eliminate the risk of including spurious noise fluctuations in the signal. After subtracting the result from the elements of $\mathbf{H}(t_j, i, n)$ within the signal borders identified in the previous step, we obtain the signal matrix $\mathbf{H}^{(sig)}(t_j, i, n)$.

The algorithm converts it to spectral reflectivity using the same formula used by Maahn and Kollias (2012) and Garcia-Benadi et al. (2020) for the MRR-2, which in turn is in agreement with the documentation provided by Metek:

$$\mathbf{SZ}_{ea}^{(lin)}(t_j, i, n) = \mathbf{H}^{(sig)}(t_j, i, n) \times \frac{c \times n^2 \times \Delta r}{\mathbf{TF}(n) \times 10^{20}} \tag{1}$$

The same formula is used for the conversion of the noise floor and standard deviation. The former is also integrated across all $m$ spectral line to derive the noise floor, $\mathbf{NF}^{(lin)}(t_j, n)$. All quantities computed so far are in linear units, $\mathrm{mm^6 m^{-3}}$.

Using $\mathbf{SZ}_{ea}^{(lin)}(t_j, i, n)$ as starting point, we can proceed with the computation of the radar variables. The first one is the attenuated equivalent reflectivity, in its logarithmic form and therefore expressed in dBZ:

$$\mathbf{Z}_{ea}^{(Proc)}(t_j, n) = 10 \times \log_{10}\left(10^{18} \times \frac{\lambda^4}{\pi^5 \times |K|^2} \sum_{i \in signal} \mathbf{SZ}_{ea}^{(lin)}(t_j, i, n)\right) \tag{2}$$

In the equation, $\lambda = 0.01238$ m is the wavelength of the instrument and $|K|^2 \simeq 0.92$ is the dielectric factor of water (Segelstein, 1981). To conclude the example of Figure 4, the spectral reflectivity computed for this specific case is displayed in panel e. Interestingly, relatively small peaks in the upper part of the spectrum may reach spectral reflectivity values comparable to much larger peaks in the lowest gate, due to the presence of the transfer function in equation 1.

Higher order moments of the spectrum give use two additional variables, the Doppler velocity and the spectral width, both expressed in $\mathrm{ms^{-1}}$:

$$\mathbf{V}^{(Proc)}(t_j, n) = \frac{\sum_{i \in signal} \mathbf{SZ}_{ea}^{(lin)}(t_j, i, n) \times vel(i)}{\sum_{i \in signal} \mathbf{SZ}_{ea}^{(lin)}(t_j, i, n)} \tag{3}$$

$$\mathbf{SW}^{(Proc)}(t_j, n) = \sqrt{\frac{\sum_{i \in signal} \mathbf{SZ}_{ea}^{(lin)}(t_j, i, n) \times (vel(i) - \mathbf{V}^{(Proc)}(t, n))^2}{\sum_{i \in signal} \mathbf{SZ}_{ea}^{(lin)}(t_j, i, n)}} \tag{4}$$

The quantity $vel(i)$ indicates the velocity associated at each spectral line number.

The last variable computed is the signal-to-noise ratio, in dB:

$$\mathbf{SNR}^{(Proc)}(t_j, n) = 10 \times \log_{10}\left(\frac{\sum_{i \in signal} \mathbf{SZ}_{ea}^{(lin)}(t_j, i, n)}{\mathbf{NF}^{(lin)}(t_j, n)}\right) \tag{5}$$

At the end, the quantities $\mathbf{Z}_{ea}^{(Proc)}(t, n)$, $\mathbf{V}^{(Proc)}(t, n)$, $\mathbf{SW}^{(Proc)}(t, n)$, $\mathbf{SNR}^{(Proc)}(t, n)$ and $\mathbf{SZ}_{ea}^{(Proc)}(t, i, n)$, together with the noise floor and level are saved in a file, at the location specified by the user in the configuration file. The optional

"quickplots" routine included in the ERUO library can provide a simple visualization of some of these products.





### 3.3 Postprocessing

Most of the parameters for the processing of raw spectra have been tuned to prioritize an increase in sensitivity, over the need to produce a clean set of variables, not contaminated by noise or interference lines. The spectrum reconstruction and the refinement of the noise level are the only two steps completely dedicated to the minimization of the impact of spurious

peaks in the raw spectra. While their contribution to the elimination of the most noticeable interference lines is remarkable, the processing output is often still affected by artifacts of lower intensity. The postprocessing aims to identify and mask these leftover non-meteorological signals.

The procedure starts by setting a minimum threshold of -20 dB on SNR, defining a new matrix of measurements to exclude, denoted by $\mathbf{EXC}(t,n)$. This quantity acts as a generic mask that can be applied over any of the radar variables: $\mathbf{EXC}(t,n) = 0$

in the regions that are left untouched by the postprocessing, $\mathbf{EXC}(t,n) = 1$ in the ones removed. The next subsections describe the two steps that modify $\mathbf{EXC}(t,n)$ in order to reach the goal of this section. These two steps are executed independently of each other. Therefore, the user can decide to execute both of them, to skip one or to ignore the whole postprocessing.

### 3.3.1 Interference line removal

The first part of the postprocessing is designed to remove the remaining interference lines, typically characterized as relatively

fixed in height and persistent for prolonged periods of time. Given the persistent nature of these lines, only range gates containing valid measurements in more than 20% of the time span of the file are considered for the analysis. By valid measurements we denote the coordinates $(t,n)$ where the radar variables computed by the processing have a not-NaN value and $\mathbf{EXC}(t,n) = 0$.

For each set of coordinates $(t_j, n_k)$, the algorithm starts by defining a window of 40 time steps around $t_j$ at $n = n_k$. This window is centered around $t_j$, with the only exception of the first and last 20 time steps in the file, where the selection becomes

asymmetrical to compensate for the borders of the matrix. We also define a set of Gaussian weights spanning the same time interval, with the maximum of the curve coinciding with $t_j$. The standard deviation of the Gaussian curve is set equal to $1/8$ of the window size.

If at least 20% of the window contains valid measurements, we count the valid measurements in an interval of 40 range gates around $n_k$, at $t = t_j$. Similarly to the previous case, the selection of range gates is symmetrical in all cases, except the

ones close to the bottom and top of the profile. A minimum threshold of 2 is imposed on the ratio between the number of valid measurements in the two windows along $t$ and $n$.

If the condition on the ratio is satisfied, the weights are summed to a temporary matrix, covering the same 40 time steps around $t_j$ at $n = n_k$. After repeating the procedure for all candidate coordinates, the resulting matrix will contain higher values at those locations where signal persists for a long time and occupies only few range gates. So, we set $\mathbf{EXC}(t,n) = 1$ in all

locations where the value of the matrix is above 20.

All numerical parameters have been chosen according to their performances in our datasets. Therefore, the user may be required to change their value in the configuration file, if the setup of the MRR-PRO undergoing the postprocessing differs from the one listed in Table 1.





### 3.3.2 Leftover noise removal

The second part of the postprocessing focuses on the sporadic small scale noise, sometimes visible in the lowest range gates in clear-sky conditions. As for the previous subsection, with valid measurements we indicate the coordinates $(t, n)$ where the radar variables computed by the processing have a not-NaN value and $\mathbf{EXC}(t, n) = 0$ (the latter retains the filtering from the previous step).

In this case, the postprocessing treats the matrix of valid measurements like an image, where each couple $(t, n)$ represents a

pixel. Using the multidimensional image processing library of Scipy (Virtanen et al., 2020), we detect all contiguous regions of valid measurements (diagonal connectivity excluded) and we count the pixels inside it. If this number is lower than 4, we set $\mathbf{EXC}(t, n) = 1$ for all coordinates $(t, n)$ inside the region.

## 4 Validation

To test the validity of the proposed method, we designed a verification phase divided in three stages. The first one ensures that

the processing and postprocessing products do not diverge excessively from the original ones (for the common detected signal) while trying to quantify the improvements in terms of sensitivity. The second one aims to provide an independent validation, testing the consistency with the variables collected by a second radar. The last one, instead, focuses on a specific phase of the processing: the spectrum reconstruction. This part of the algorithm is arguably the one that diverges the most from libraries presented in other studies for the processing of similar radars, such as the MRR-2. Therefore, we deemed necessary a dedicated

verification, to ensure that the reconstruction indeed improves the quality of the products.

### 4.1 Comparison with initial MRR-PRO products

The comparison between the ERUO products and the original variables, extracted directly from the MRR-PRO data files, is performed over the four datasets presented in Section 2. Given the issue in the transfer function experienced by the MRR-PRO 23 during the two campaigns, particular care will be taken in comparing its results with the ones derived from the other

two datasets.

### 4.1.1 Sensitivity curves

To highlight the improvement in sensitivity, we computed the 2-dimensional histogram of attenuated equivalent reflectivity factor and range gate number. For each of the four datasets, three histograms have been produced: the first one using $Z_{ea}$ from the original MRR-PRO files, the second and third ones using the ERUO processing and postprocessing output, respectively.

We extracted the minimum value and the quantile 0.01 from the empirical distribution of $Z_{ea}$ at each range gate. These quantities are interpreted as approximate indicators of the minimum detectable signal at each height. The difference between the set of statistics computed from the original variables and the ones derived from the processed (or postprocessed) ERUO products provides us with an estimate of the improvements in sensitivity that our algorithm is able to deliver.


As a summary of the typical behavior observable in such histograms, we decided to display in Figure 5 the comparison
between the original and postprocessed measurements for the MRR-PRO 06 and MRR-PRO 22 datasets. These two radars
were chosen because their measurements are unaffected by the issue in the transfer function observed for the MRR-PRO 23,
which results in a fairer comparison.

The effects of interference lines on a dataset is clearly visible in both panels, where small, isolated clusters of bins with
relatively high counts can be seen starting from an height of around 3 km. In panel b, most of the effect of interference is not
visible anymore, except for the strongest one, at approximately 4.5 km, and a smaller one 1 km below. Additionally, the count
associated to their bins is significantly lower than the one seen in panel a.

In panel d the interference clusters are accompanied by a large amount of noise, characterized by relatively low counts and
reflectivity spanning a small interval close to the minimum detectable values. Even though the position of the interference is
similar between the two datasets, their appearance in the spectra differ significantly (see subsection 4.3), and ERUO is able to
produce a cleaner result for the MRR-PRO 22 dataset.

In each of the mentioned four panels, two sets of dots are visible: the blue ones denote the minimum of the empirical
distribution of $Z_{ea}$ at each range gate, while the green ones show the quantile 0.01. Panels c and f display the difference
between each set of statistics for the two campaigns, with a positive value indicating that the ERUO products reach a lower
$Z_{ea}$ at that specific range gate. We can observe how the ERUO processing improves the sensitivity of more than 10 dBZ
in the lowest part of the profile, up to approximately 2 km above the first gate. Above that, the improvement is gradually
reduced, reaching 0 dBZ at an height between 2.1 and 2.5 km, depending on the MRR-PRO model and the statistic considered.
Both datasets have been collected at PEA, where precipitation events were relatively faint and shallow, resulting in a lack of
measurements at higher altitudes. The few isolated values above that height, visible in panel c, can be safely ignored, since
they refer to the interference lines visible in both products. Lower values of the two statistics may actually indicate that those
interference lines are fainter in the ERUO postprocessed outputs.

The same set of statistics computed over all datasets are displayed in Figure 6, with the results for the ERUO processing
output in the top row of panels and postprocessing ones on the bottom row. Note that the transfer function reconstruction
radically affects the MRR-PRO 23 measurements, and a direct comparison of its sensitivity with the ones of the other two
MRR-PRO is not possible.

At first, the sensitivity gain after the processing appears to be the largest for all datasets. However, we suspect that most
of the gain is caused by the noise that appears in the lower sections of the profile, particularly marked when the antenna is
covered in snow. The postprocessing removes this noise, resulting in a straighter profile. The sensitivity improvements remain
large, and for the MRR-PRO 06 and MRR-PRO 22 it appears to be between 10 dBZ and 15 dBZ depending on the statistic
considered, at least in the first 2 km of the profile. For the MRR-PRO 23 and ICEGENESIS dataset the improvements appear
larger, between 10 dBZ and 20 dBZ. These higher gains are most probably due to the change in transfer function, since the
reconstruction results in a higher $\mathbf{TF}(n)$ for many range gates in the upper part of the profile. Finally, in the lowest range
gates we always observe a lower minimum detectable reflectivity for the original products. However, a closer inspection of the
time series (such as the one discussed in subsection 4.2) reveals that $\mathbf{Z_{ea}}(t, n)$ usually experiences a drop at low $n$, much less





marked in the ERUO products. This drop may be caused by an incorrect noise floor estimation in the original algorithm by
Metek, probably linked with the low raw spectra values typical of this region of the spectrum (see panel a of Figure 2).

### 4.1.2 Direct comparison of radar variables

In this subsection we check how consistent the postprocessed ERUO products are with the variables already available in the
original MRR-PRO files.

Given the different issues affecting each dataset, we can expect that a variety of behaviors can be observed in the comparison,
as exemplified by Figure 7. As for the sensitivity estimate, we decided that the MRR-PRO 06 and MRR-PRO 22 datasets are
the best choice for a fair comparison, given the absence of issues in the transfer function.

The 2-dimensional histogram of panel b offers the most direct comparison, given the better removal of interference by ERUO
for the MRR-PRO 22 dataset compared to the MRR-PRO 06 one. For high reflectivity values, the central peak of the histogram
is relatively well aligned with the green line, which in turn is parallel to the identity line. At lower $Z_{ea}$, we observe a small
drift from this line, with the peak following more closely the linear fit, and larger spread of the values. The MRR-PRO 22 was
deployed at the onset of the Antarctic plateau, at the highest altitude among the three radars at PEA, and therefore observed the
faintest and most shallow precipitation events. We suspect that the different behavior of the two algorithms at the lowest range
gates, where the raw spectrum drops suddenly, may be among the causes for the observed deviation of the histogram peak
from the identity line. The MRR-PRO 22 dataset was more heavily affected by noise than the other datasets, as partially visible
in panel d of Figure 5. Even though only the noise in the upper gates is visible in this panel, a visual inspection of the $Z_{ea}$
time-series reveals that noise is often recorded in clear sky situations. By analyzing the ICEGENESIS dataset, during which
we have reliable co-located observations of precipitation, we noticed that the noise is more noticeable after snowfall events,
with the antenna still covered in snow due to the absence of heating. The same may hold true for the MRR-PRO 23 dataset at
PEA, even though we cannot confirm it since we lack independent precipitation measurements at the site.

The comparison for the MRR-PRO 06, shown in panel e, is instead plagued by the the numerous interference lines, visible
in the upper right corner of the plot. Small peaks associated with them can be noticed in the two empirical distribution of
$Z_{ea}$ in panels d and f. In agreement with what we saw in panel a of Figure 5, interference lines not removed by ERUO still
appear fainter than in their original counterpart. This explains their position in panel b of Figure 7, always below the identity
line. Additionally, a large spread can be observed in lowest reflectivity values, similarly to the MRR-PRO 22 case. Overall, the
alignment of the central peak of the 2-dimensional histogram with the identity line is better than in the MRR-PRO 06 case, as
confirmed by the significant overlap of the green (median difference) and blue (linear fit) lines.

A similar comparison can be repeated for the remaining two variables ($V$ and $SW$). For completeness, we also included the
other two datasets in the analysis, even though the issue with the transfer function necessarily results in enlarged differences
for $Z_{ea}$. To summarize the results, we decided to compute three statistics from each comparison, each shown in a different row
of Figure 8: the median difference (original MRR-PRO products minus the ERUO ones) in panels a, b and c, the interquartile
range in panels d, e and f, and the Pearson correlation coefficient in panels g,h and i.





From panel a we can estimate an offset between the ERUO and the original MRR-PRO products: around 1 dBZ in normal conditions, and up to approximately 4 dBZ in presence of issue with the transfer function. A part of the offset may be explained by the difference in the noise floor detection between ERUO and the algorithm used by Metek. We suspect that leftover interference lines may also be a contributor to this difference, due to their consistently lower reflectivity values after the ERUO processing and postprocessing, as shown by the sensitivity plots.

Among the Doppler velocity offset, the only noteworthy one is the ICEGENESIS one, equal to -18.1 m/s. The original MRR-PRO algorithm had problems in computing the variable, as confirmed in the next subsection, where the comparison with WProf is presented. The behavior may be similar to a problem with the Doppler velocity field described in Garcia-Benadi et al. (2020) for the MRR-2. Note that the Doppler velocity computation is, in theory, unaffected by the transfer function issue, which results in a better agreement, at least for the MRR-PRO 23 dataset.

More significant differences can be seen for the spectral width, in panel c. We suspect that it may be due to differences in signal detection: the higher sensitivity of ERUO may result in a lower noise level, which may cause the peak associated with the detected signal to appear wider, and therefore have a larger $SW$.

Panel d can be explained similarly to panel a, with the interference lines and noise playing a significant role in widening the spread of points around the identity line of the relative scatterplots. Panel e, instead, may be misleading. While the IQR of the differences seems small, particularly large values can be found at the extremes of the empirical distribution of the differences. Due to the absence of an explicit antialiasing in the algorithm used by Metek for producing the original MRR-PRO products, several of the original $V$ values are approximately $v_{ny}$ apart from their ERUO counterpart.

The values of Pearson correlation coefficient are close to 0.9 for the MRR-PRO 06 and MRR-PRO 22, as expected from the high level of agreement shown in Figure 7. The transfer function differences negatively impact the correlation, especially for the MRR-PRO 23. While antialiasing may not affect directly the quantiles 0.25 and 0.75 used for panel e of Figure 8, its influence can be seen clearly in panel h. The correlation coefficient is significantly lower than in panel g for all datasets. While the ICEGENESIS case may be explained mostly by the -18.1 m/s shift in the original MRR-PRO values, the other datasets are still heavily affected by the aliasing, which aligns the $V$ couples in an hypothetical scatterplot on three separate lines, parallel to the identity line.

## 4.2 Comparison with WProf

Given the presence of WProf alongside the MRR-PRO during the ICEGENESIS field campaign, we have access to an independent dataset for the verification of the ERUO products. Unfortunately, the MRR-PRO deployed at the site was affected by the issue in the transfer function previously described in subsection 3.2.1. Any improvement in the agreement with WProf will be partially due to its reconstruction. Therefore, this section is not intended as an estimate of the improvements of the ERUO products compared to the original MRR-PRO variables, but as an independent verification of the validity of the signal recovered by the proposed method.

Since the two radars record data at a different temporal and vertical resolution, we decided to remap the measures of WProf on the MRR-PRO resolutions, which are the coarsest. An example of the remapped data, alongside the two sets of MRR-PRO





measurements, can be seen in Figure 9. In panel c we can immediately spot the interference line typical of the ICEGENESIS dataset, just above the 1 km height mark, of which we can only see some traces in the ERUO products of panel b. The difference of sensitivity between the two panels is also evident, especially at higher altitude. A third issue affects the profiles of panel c: $Z_{ea}$ increases with height much faster than the WProf measurements of panel a. This behavior is particularly evident after

15:45, when the MRR-PRO measures an increase of reflectivity with height above 1.5 km, while WProf sees a decrease. A discussion of this effect and its possible causes is presented in Appendix-A.

Before extending the comparison of $Z_{ea}$ to the whole ICEGENESIS campaign, we need to define clearly what measurements are suitable for it. Given the difference in frequency, we can expect that measurements in rainfall would be particularly attenuated for WProf, therefore we decided to manually exclude the only rain event recorded in the period considered from

the comparison. Additionally, large hydrometeors can be problematic, since we could have cases in which the measurements at W-band are dominated by Mie scattering, while the Rayleigh scattering is still valid at K-band. This would artificially increase in the difference between the $Z_{ea}$ measured at the two bands, spoiling the comparison. Given the presence of MXPol at few kilometers of distance from the site, we could use the results from both the hydrometeor classification (Besic et al., 2016) and the demixing algorithm (Besic et al., 2018) to extract profiles directly above the MRR-PRO and WProf site, giving

us information on both the dominant hydrometeor type and the proportion of each hydrometeor class. After remapping them on the same MRR-PRO temporal and spatial resolution mentioned before, we decided to use this information to select which radar volumes are suitable for the comparison. The first rule we defined is the most relaxed one: only volumes dominated by crystals, according to the classification results of Besic et al. (2016), will be used for the comparison between the MRR-PRO and WProf. Crystals are among the most common hydrometeor type in our dataset, and they are the smallest in size, therefore

the most unlikely to cause an increase in the difference between the two bands. Naturally, we can expect some crystals to be particularly large and still become Mie scatterers at W-band, but this would still be less common than with other hydrometeor classes with larger diameters. A second comparison is then performed on a stricter set of rule: in addition to enforcing a majority of crystals in the volumes, we also required that less than 20% of the hydrometeors in it are aggregates, using the percentages provided by Besic et al. (2018). Aggregates are also common in our measurements and they can reach sizes

considerably larger than crystals. As can be seen in Kneifel et al. (2015), the difference between the logarithmic difference of reflectivity values between X-band and W-band deviates the most from linearity when considering aggregates. This deviation is caused by aggregates becoming Mie scatterers at W-band, a fact that would hold true also in our comparison, even though our comparison involves K-band instead of X-band.

Two 2-dimensional histograms obtained using this second set of rules are shown in Figure 10. $\mathbf{Z}_{ea}^{W}(t,n)$ is compared with

$\mathbf{Z}_{ea}(t,n)$ (panel b) and with the ERUO postprocessed products (panel d). By comparing the two we can immediately spot the effect of interference, causing a cluster in the top left corner of panel b, less marked in panel e. Similarly to the previous comparison, we can also observe how the ERUO output reaches lower $Z_{ea}$ values. However, the most striking feature is the angle between the central ridge of higher counts in the 2-dimensional histogram and the identity line, visible in panels b and much less noticeable in panel e. For the original MRR-PRO products this is obviously the effect of the transfer function issue.

For panel e, instead, it appears that the central peak in the 2-dimensional histogram is approximately aligned with the green





curve, parallel to the identity line. However, the spread remains large, with ERUO recording a particularly high number of reflectively values lower than their WProf counterpart. One possible explanation is the noise in the MRR-PRO measurements, in particular during precipitation events too faint to be recorded by the latter and strong enough to be detected by WProf. A second contributing factor could be the artificial lowering of $Z_{ea}$ at the lowest $n$. The phenomenon is more marked in the

original MRR-PRO data, but it is not completely solved by ERUO. The stronger attenuation at W-band compared to k-band can also be a contributing factor.

Similarly to the comparison presented in the previous subsection, we decided to summarize the results of the comparison for each variable in Figure 11. ERUO appears to perform better than the original products in all panels. However, as explained at the beginning of this subsection, a significant part of the improvement for $Z_{ea}$ can be attributed to the transfer function

reconstruction. Additionally, in the case of panel a, the exact value of the median difference loses importance when we are unsure on the contribution of calibration differences on this offset. However, the improvements in interquartile range and correlation can, at least partially, be explained by the reduction of the contribution of interference lines. Intuitively, the latter appear in the 2-dimensional histogram as a cluster of high $Z_{ea}$ for the MRR-PRO and low $Z_{ea}$ for WProf, which enlarges the spread and worsens the overall correlation.

On the contrary, differences in Doppler velocity cannot be linked as directly to the transfer function, since the variable depends mostly on the position of the recorded signal in the velocity range. Panel b shows that the ERUO products have a significantly smaller median difference from $\mathbf{V}^W(t, n)$. This is obviously linked with the issue with the Doppler velocity in the original MRR-PRO files previously mentioned. We observe a decrease of the interquartile range of the difference with WProf also for $V$, probably caused by the antialiasing.

Overall, even though a direct comparison of the original and ERUO products for this dataset is not completely fair, the improvement in the series of statistics displayed in Figure 11 still confirms that ERUO is recovering valid measurements. As hinted by Figure 9, many measurements that were below the sensitivity in the original products are now present in the ERUO ones. The improved agreement with WProf suggests that those newly visible signals are indeed valid precipitation measurements and they brings valid information, previously unavailable, to the dataset.

## 4.3   Performance of the spectrum reconstruction

The spectrum reconstruction plays a crucial role in creating a clean output, eliminating the strongest interference before they can affect any of the variables. In all four datasets, interferences appear as spurious peaks in raw spectrum anomaly ($\mathbf{A}(i, n)$). These peaks cover any return of lower intensity and they are masked by stronger signals. Since we cannot know what the real meteorological spectrum would have looked like without the interference, we cannot directly test the performances of our

reconstruction.

However, we can extract some of these spurious peaks in $\mathbf{A}(i, n)$ and apply them to unaffected regions of the spectra from our datasets. Then, we can apply the spectrum reconstruction to these modified spectra, process them with ERUO and compare outputs with the respective unaffected ones. If we process the modified spectra without first reconstructing them, we can also obtain a baseline that allows us to estimate the improvement brought by the reconstruction.



As starting point, we extracted 180 examples of raw spectrum anomalies associated to interference lines, divided equally among the four datasets. We used the previously defined $\mathbf{IM}(i,n)$ to define the extent of each anomaly. Four of them can be seen in Figure 12, in which each panel shows the typical shape of the spurious peaks in $\mathbf{A}(i,n)$ for its respective dataset. Their exact position changes during the time span of the campaign, so the region extracted is usually larger than the main anomaly peak. These four interference types have been chosen to represent as much as possible the variety that we observed.

Therefore, even though the MRR-PRO used for the ICEGENESIS campaigns is the same MRR-PRO 23 deployed at PEA, different interference lines have been chosen to represent the two datasets. Additionally, since the anomaly in panel a is not symmetrical with respect to the vertical axis, we flipped half of them. As a consequence, in the flipped anomalies the rightmost peak becomes closer to lower velocity values, which are more likely to contain precipitation.

    Since the extracted interferences are stored in a $i = 32 \times n = 32$ grid, we had to identify a region of similar size in each
dataset in which $\mathbf{IM}(i,n) = 0$. The chosen central gates are: $n = 32$ for the MRR-PRO 6, $n = 73$ for the MRR-PRO 22 and $n = 125$ the MRR-PRO 23. This selection allows for extra 5 range gates on each side of the region, to avoid being too close to already existing interference lines. In addition to the constraints imposed by $\mathbf{IM}(i,n)$, the central gates have been spaced out through the profile to monitor the effect of interference at different heights.

    An example of how the extracted anomalies are applied to a spectrum and of the subsequent reconstruction is shown in
Figure 13. Since the reconstruction acts on anomalies ($\mathbf{A}(i,n)$) and not directly on the raw spectrum, we decided to display the former in the example figure. In panel a we highlighted the region to which the interference will be applied by delimiting its border with two dashed lines. As expected, the meteorological signal in the region seems unaffected by any noticeable spurious peak. In panel b, one of the anomaly matrices extracted from the ICEGENESIS dataset has been applied to the profile. At each sets of coordinates $(i,n)$, the resulting value has been chosen as the maximum between the original $\mathbf{A}(i,n)$ and the extracted
interference anomaly. As a result, it is possible to see both the meteorological signal and two interference lines of different intensity. The spectrum reconstruction takes the resulting matrix as input, and masks only a fraction of the entries, as shown in panel c, following the rules detailed in subsection 3.2.2. Finally, the reconstructed anomaly is show in the last panel. This output undergoes the standard processing, from which we derive $Z_{ea}$, $V$ and $SW$. The whole operation has been repeated for 120 profiles, divided equally among the three datasets collected at PEA.

The variables derived from the reconstructed and altered profiles are compared with the ones computed from the original, unaltered raw spectra. The result of the comparison are shown in Figure 14. The criteria used for the comparison are: the mean absolute error (MEA) in the first row of panels, the root mean square error (RMSE) in the second, and the Pearson correlation coefficient (PCC) in the third. All comparison criteria are computed only over the range gates covered by newly added anomalies. Since the reconstruction often only affects a small fraction of these gates, the median difference could not be
used to estimate the offset (or error, in this case) as done in the previous subsections, since its value was almost always null.

    The value of the three statistics changes drastically between the three MRR-PRO. Given the different central height at which the anomalies have been applied, the difference may be due to the relative scarcity of precipitation data at higher altitudes. To compute each of the three statistics, we need that a signal is detected, at least for a fraction of the range gates considered, in all





three profiles: the reconstructed one, the modified one and the unaltered one. Therefore, the comparison shown in the plot only

takes in account precipitation data.

It should be noted that the overlapping of meteorological signal and interference is the most complex situation to reconstruct for ERUO, therefore the skills discussed here represent a "worst case scenario". As exemplified by Figure 9, the traces of the original position of the interference line in the ERUO products can often be seen only over precipitation measurements, and not in the clear sky sections.

In all cases the reconstruction reduces the error on the radar variables, while improving their correlation with the ones derived from the unaltered profiles. However, both MAE and RMSE indicate that the variables differ considerably from their ideal value. By observing simple example of reconstruction, we could identify two critical situations that are not handled correctly by ERUO. The first one is the overlapping of precipitation signal with interference in the shape shown in panel a of Figure 12. As described in subsection 3.2.2, the algorithm checks whether the interference is an isolated peak, or a line spanning

the whole range $i = 1, .., m$. When one of the two peaks shown in panel a of Figure 12 intersect the precipitation signal, it falls in neither of the two categories, and therefore it is not reconstructed. The second case is caused by weak interference, such as the one in panel b of Figure 12, or the bottom line in the anomaly displayed in panel d. As can be seen in panel c of Figure 13, sometimes the anomaly is below the threshold used for masking it, and therefore is never flagged for reconstruction. This behavior can be controlled by the user by setting lower thresholds in the configuration file. However, this may lead to an

increase in false positive when flagging the pixels to reconstruct.

## 5   Summary and conclusions

In this study we presented the ERUO library, a new processing technique for the MRR-PRO raw spectra, developed specifically for measurements in snowfall. The method has the two-fold objective of minimizing the effect of a series of issues affecting the raw spectra, such as interference lines, and improving the quality of the output radar variables. The algorithm is divided in

three stages, each with its own set of configurable parameters that can be tuned by the user to better fit their measurements.

The first step, the preprocessing, uses information accumulated over an extended period of time to extract some quantities used in the correction of the raw spectra. In particular, the quantity $\mathrm{BM}(i,n)$, which compensates for the power drop at the extremes of the Doppler velocity range, is computed at this stage. Previous scientific work used interpolation to solve an analogous issue with the MRR-2 spectra, but this could lead to underestimation of the signal is case of aliasing (Maahn

and Kollias, 2012). The ERUO preprocessing, instead, is able to make use of the measurements of the instrument by simply adding back an offset purely derived from the dataset itself. On the other hand, the preprocessing contains the first of the major limitations of the ERUO library. Given the need to load all available data at once to extract the median spectra, large datasets (due to high frequency of acquisition or large temporal duration of the measurement campaign) may be problematic to process. While this particular problem can be circumvented as described in subsection 3.1.3, the proposed solution still

requires a manual intervention of the user.



The next step, the processing, computes a series of variables using the preprocessing outputs and the raw spectra as starting point. A crucial point during this procedure is the spectrum reconstruction. The $\mathbf{IM}(i,n)$ matrix computed during the previous stage indicates region of the spectra likely to contain interference. A series of conditions aims to isolate only the location of the interference peak, and the meteorological signal that has been covered by it is estimated from the surrounding unaffected spectra. In the verification stage, particular attention has been dedicated to this reconstruction. We tested its performances by applying real interference anomalies from all our datasets to unaffected regions of the spectra collected by three MRR-PRO. The reconstruction results in variables closer to the ones obtained from the unaltered spectra, when compared to their not-reconstructed counterparts. Better skills are obtained for interference lines placed at higher $n$ values, hinting that the reconstruction may have more difficulties when precipitation and interference coexist in the same profile. A visual inspection of the MRR-PRO 06 reconstructed products shows that the reconstruction fails when isolated peaks, like the ones shown in panel a of Figure 12, partially overlap with meteorological signal. This represents one of the biggest limitations of the proposed method. A possible future solution may involve the modification of the spectrum reconstruction routine, by changing the conditions for interference in the form of isolated peaks.

The final product of the processing are the variables $Z_{ea}$, $V$, $SW$ and $SNR$. The proposed noise level results into large improvements in terms of sensitivity in normal condition, estimated to be between 10 dBZ and 15 dBZ, depending on the statistics used. The improvement seems to taper off at about 2.0 km above the first range gate, even though it is unclear if this is caused by the drop of precipitation measurements at those heights in our datasets. The processing also involves a simple antialiasing, based on the vertical continuity of the line connecting the local maxima of the spectrum at each range gate.

The MRR-PRO 23, source of two of the datasets used in the analysis, presents an abrupt cut-off in the transfer function at $n = 128$. The ERUO library is able to detect the issue and the $\mathbf{TF}(n)$ value is re-sampled over the correct number of range gates. This allows for the recovery of measurements in the upper part of the profiles, previously missing in the variables available in the original MRR-PRO data.

The last step of the algorithm is the postprocessing, that eliminates leftover interference lines and some isolated noise, often present in the lowest gates when the antenna is covered in snow. Both processing and postprocessing outputs are first compared with the original MRR-PRO variables over the four datasets. Given the transfer function issue of the MRR-PRO 23, a direct comparison of $Z_{ea}$ is meaningful only for the two datasets collected at PEA by the other MRR-PRO. For the $Z_{ea}$ in these two datasets we observe the best agreement, with a median difference always within 1 dBZ and a Pearson correlation coefficient close or above 0.95. Instead, $V$ and $SW$ are more affected by the lack of antialiasing in the original MRR-PRO processing algorithm. In this regard, ICEGENESIS differs from the other datasets, achieving worst performances due to an issue in the velocity computation of the original MRR-PRO processing algorithm resulting into a -18.1 m/s offset. Overall, excluding these sources of spurious difference and the impact of the transfer function issue on $Z_{ea}$, it appears that ERUO products agree reasonably well with the original variables.

A comparison of the $Z_{ea}$ and $V$ measurements with a second radar, WProf, has also been performed. Only the ICEGENESIS dataset has been used in this case, since it is the only one in which the two radars were co-located. Unfortunately, MRR-PRO 23 was used for this campaign, and its measurements are affected by the transfer function issue. The analysis has been repeated





for two different sets of conditions, imposing first a condition on the sole dominant hydrometeor type (ice crystals) and then also on the maximum proportion of aggregates (20%). In both cases the ERUO products have shown a lower median and IQR of the difference between the two radars, and a higher Pearson correlation coefficient. The same velocity computation issue mentioned before also led to a significantly reduced median difference of the ERUO product from the WProf counterpart,

when compared with the original MRR-PRO velocity. For $Z_{ea}$, however, it is impossible to distinguish exactly the effect of the transfer function reconstruction from any other improvements brought by the ERUO processing. The original truncated transfer function results in a reflectivity profiles that increases with height much faster than WProf, while the difference is much more stable when the reconstructed one is used. This implies that the results of the comparison can be used as independent verification of the validity of the ERUO products, but not as fair comparison between the latter and the original MRR-PRO

variables. Overall, the agreement between the ERUO variables and the ones collected by WProf suggests that the former is recovering valid measurements.

Finally, an achievement of the ERUO processing that may be hard to quantify is the enhancement in readability of the precipitation structure in time-height plots. ERUO is able to recover precipitation signals that were too faint to be recorded in the original data, especially at higher altitudes. As can be seen in Figure 9, the removal of the main interference line and the

improvement in sensitivity result in a much clearer series of profile, in which the structure of the precipitation system can be more easily be understood.

*Code availability.* The ERUO library is available for download at https://github.com/alfonso-ferrone/ERUO.git

## Appendix A: Height dependence of the offset in the ICEGENESIS dataset

An inspection of the time-series of attenuated equivalent reflectivity factor recorded by the MRR-PRO and by WProf reveals

some differences in the vertical trends of $Z_{ea}$. While the original MRR-PRO variable increases much faster than its WProf counterpart, the ERUO products appears to be more in agreement with the W-band radar.

Therefore, we decided to investigate the $Z_{ea}$ comparison more closely. By studying the change of the median difference value with altitude, we were able to quantify the discrepancy in behavior between the two sets of MRR-PRO products. As can be seen in panel a of Figure A1, the difference between the WProf and original MRR-PRO attenuated reflectivity factor

decreases steadily until the height of approximately 1 km, after which it stabilizes. The interquartile range of the difference also experiences a decrease, even though its value is significantly smaller than the variation in the median value. On the contrary, the median difference for the ERUO products does not experience a significant drift with height, and the value appears to oscillate around 2 dBZ for the whole profile. A less marked tapering of the IQR can also be seen in the upper range gates. We suspect that the IQR remains larger than for the original MRR-PRO case because a higher amount of measurements are recovered by

ERUO at those heights.





The discrepancy between the two sets of products can be explained by looking at the transfer function before and after the reconstruction, described in subsection 3.2.1 and illustrated by Figure 3. Since the original $\mathbf{TF}(n)$, used by the MRR-PRO to computed the variables available in the initial data files, is usually lower than the reconstructed one for high $n$, the original spectral reflectivity is necessarily higher than the one derived by ERUO. This results in an artificial increase of $Z_{ea}$ with height for the original products, which is absent in ERUO thanks to the reconstruction.

The Pearson correlation coefficient, in panel b, shows a better agreement between the ERUO and original products. Both experience a decrease with height, followed by a small rise in value for the last 500 m of the profile. Only minor differences exist between the original $Z_{ea}$ values and the ERUO ones, with the latter slightly outperforming the former in the range 1 km to 2 km. Overall, the transfer function issue has only a limited impact on correlation, when smaller section of the profiles are analyzed independently from each other.

*Author contributions.* A.F. and A.B. designed the study, with inputs from A.-C.B.-R. on the verification section. A.F. and A.B. deployed and maintained the instruments at PEA. A.-C.B.-R. and A.B. deployed and maintained the instruments for ICE GENESIS. A.F. processed the data for PEA, A.-C.B.-R. processed the ones for ICE GENESIS, including executing the hydrometeor classification and demixing algorithm for the MXPol data. A.F. prepared the manuscript with contributions from A.-C.B.-R. and A.B., and supervision from A.B. All authors have read and agreed to the published version of the manuscript.

*Competing interests.* A.F and A.-C.B.-R declare that no competing interests are present. A.B is a member of the editorial board of the journal (associate editor).

*Acknowledgements.* We would like to thank all the EPFL-LTE collaborators involved in the deployment and management of the radars during the field campaigns at PEA and ICEGENESIS. We are grateful to all the personnel of the PEA station, including management, technicians, field guides and everyone at the station that made the campaign possible. We are also thankful to the personnel of the airport of La Chaux-de-Fonds for their assistance in various stages of the installation and removal of the instruments. Finally, we greatly appreciated the innumerable suggestions of Josué Gehring, Gionata Ghiggi and Monika Feldmann, which aided us in shaping the algorithm and the subsequent verification.





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



**Table 1.** Measurements setup of all the MRR-PRO used in the study. A symbol has been associated to each parameter, to ease the referencing in the text.

| Parameter name | Parameter symbol | Value |
|---|---|---|
| Number of range gates | $n_{max}$ | 256 |
| Number of lines in spectrum | $m$ | 32 |
| Time of incoherent averaging | $\Delta t$ | 10 s |
| Range resolution | $\Delta r$ | 25 m |
| Height range | $hr$ | 6.4 km |
| Velocity resolution | $\Delta v$ | 0.19 ms$^{-1}$ |
| Nyquist velocity range | $v_{ny}$ | 6.0 ms$^{-1}$ |



**Table 2.** Measurements setup of WProf during the ICEGENESIS campaign.

| Chirp number | Vertical range [m] | Vertical resolution [m] | Nyquist velocity [m/s] | Velocity resolution [m/s] |
|:---:|:---:|:---:|:---:|:---:|
| 1 | 104 - 894 | 7.45 | 10.95 | 0.021 |
| 2 | 905 - 3796 | 32 | 6.93 | 0.027 |
| 3 | 3805 - 9984 | 32 | 3.2 | 0.0127 |





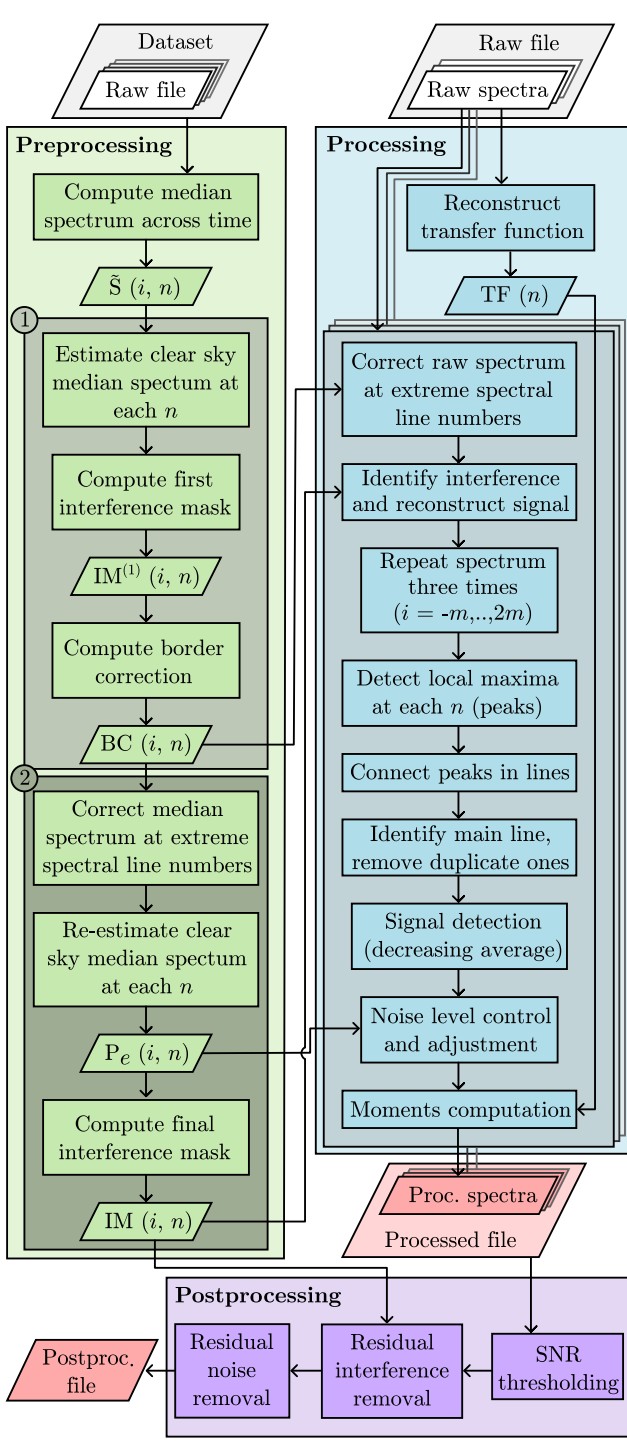

**Figure 1.** Flowchart of the ERUO algorithm. Romboid shapes indicate inputs (white background) and outputs (red backgrounds), while rectangles indicate processing steps. The numbers 1 and 2 at the top-left corner of the two darker green boxes inside the preprocessing denote the two iterations described in subsection 3.1.



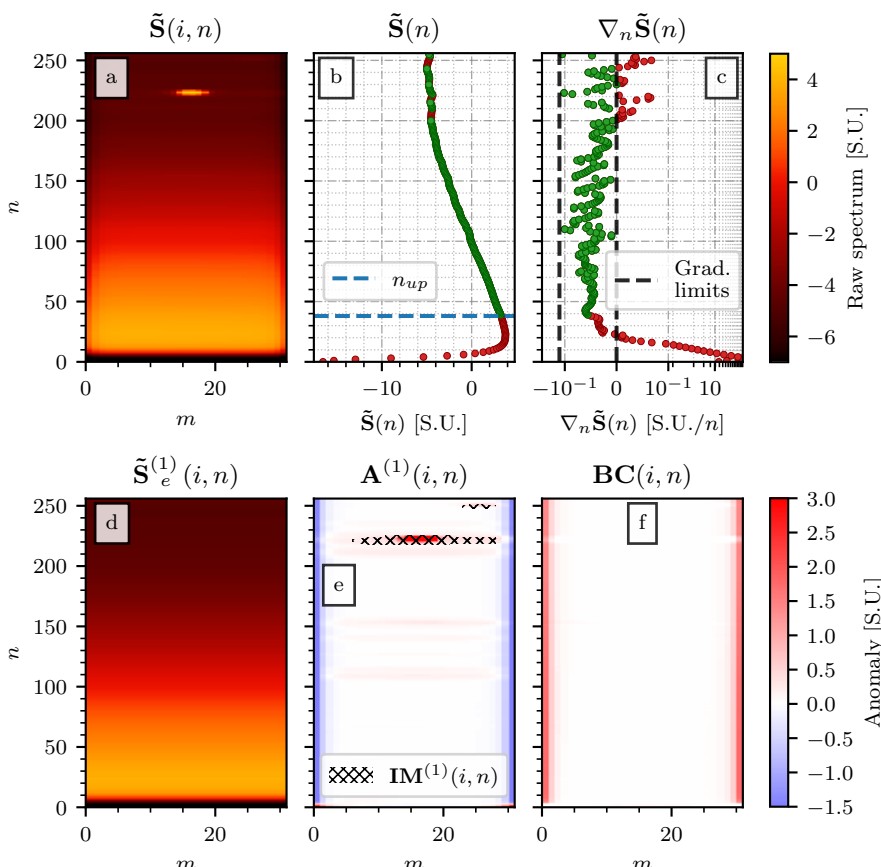

**Figure 2.** First iteration of the preprocessing of the MRR-PRO 06 dataset. Panel a shows $\tilde{\mathbf{S}}(i,n)$. Panel b displays its median value at each range gate ($\tilde{\mathbf{S}}(n)$) with the entries used in the fit of the median profile colored in green, and the excluded ones in red, with the dashed blue line at $n = n_{up}$ delimiting the upper region. Panel c shows the vertical gradient of $\tilde{\mathbf{S}}(n)$, with the same color coding as panel b. The vertical, black dashed lines denote the limits for accepted gradient values. The reconstructed median profile obtained by tiling $m$ times $\mathbf{P}_e^{(1)}(n)$ is shown in panel d. Panel e shows the anomaly from this baseline profile ($\mathbf{A}^{(1)}(i,n)$), with a superimposed hatching representing the first estimated interference mask ($\mathbf{IM}^{(1)}(i,n)$). Finally, the last panel displays the first product of the preprocessing, $\mathbf{BC}(i,n)$.





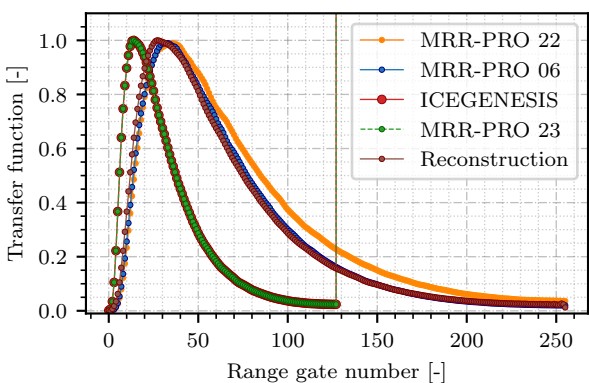

**Figure 3.** Transfer function from all the datasets used in the study. Colors are assigned to each dataset as follows: blue for MRR-PRO 06, orange for for MRR-PRO 22, green for for MRR-PRO 23 and red for ICEGENESIS. Since the MRR-PRO used during ICE GENESIS is the same MRR-PRO 23 deployed at PEA, their transfer function is the same. The output of the transfer function reconstruction for these two last datasets is shown by the dotted brown line.



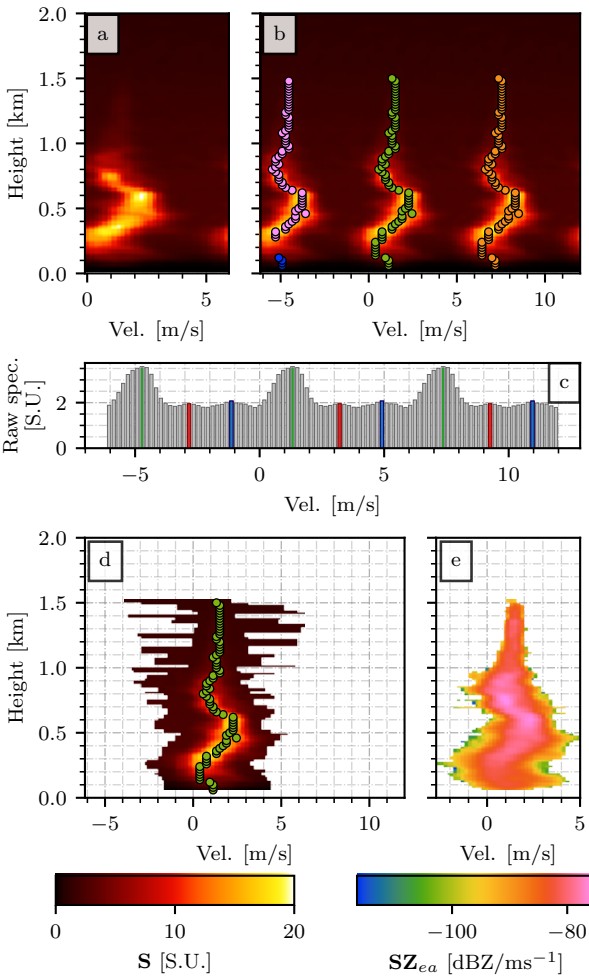

**Figure 4.** Key steps of the processing. Using the raw specta recorded on December 18, 2021, at 03:07:30 UTC by the MRR-PRO 06 as the starting, panel a shows the result of the spectrum reconstruction and the correction at the extreme Doppler velocities. Panel b displays the local maxima identified at each range gate, after they have been connected in lines. A different color has been assigned to each line. Due to aliasing, two different lines (blue and pink) have been assigned to the leftmost precipitation signal. Panel c shows the intermediate steps in the peak detection for $n = 50$. The gray bars represent the raw spectrum, the red ones are the entries identified as peaks but rejected by the prominence threshold, the blue ones are peaks rejected by the relative prominence threshold, and the green ones are the accepted ones. Panel d displays the largest $m$ entries selected around each peak in the main line. Finally, panel e illustrates the product of the signal identification and conversion in spectral reflectivity. The maximum height has been limited to 2.0 km, since no precipitation was recorded above this altitude. The colorbar at the bottom left refers to panel a, b and d, while the one on the right to panel e.

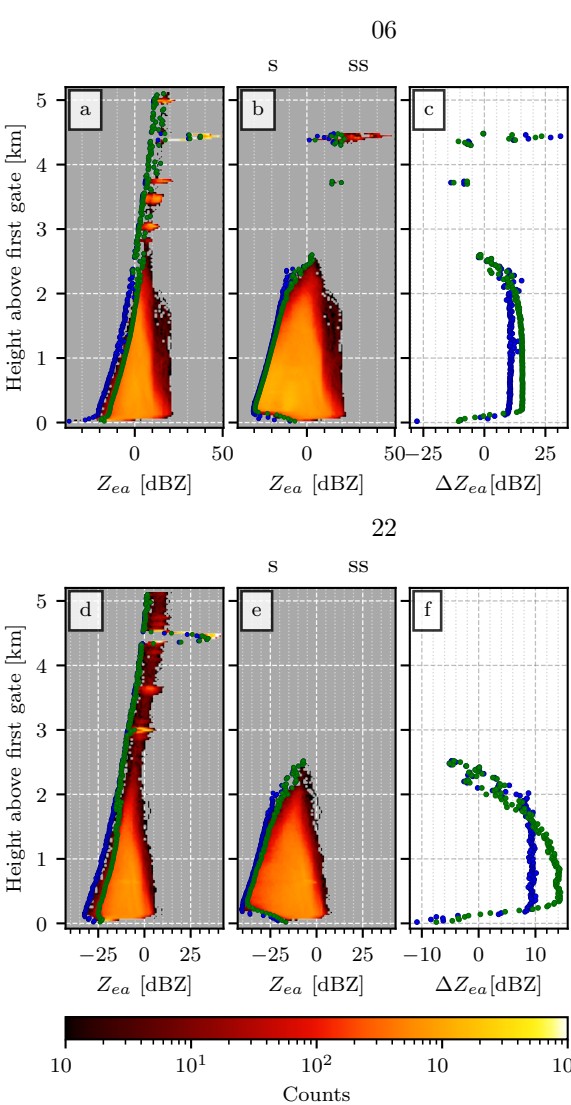

**Figure 5.** Sensitivity curve for the MRR-PRO 06 (panels a, b, c) and MRR-PRO 22 (panels d, e, f) datasets. Panels a and d show a 2-dimensional histogram of the original attenuated equivalent reflectivity factor, while b and e display the postprocessed one. The coloring of the bins of all histograms follow the same color scale, shown at the bottom of the figure. The blue and green dots highlight respectively the minimum and the quantile 0.01 of the $Z_{ea}$ empirical distribution at each range gate. The differences between the two series of values, computed by subtracting the statistics of the ERUO postprocessed products from the ones of the original MRR-PRO variables, are displayed in panels c and f, following the same color convention as the first four panels.



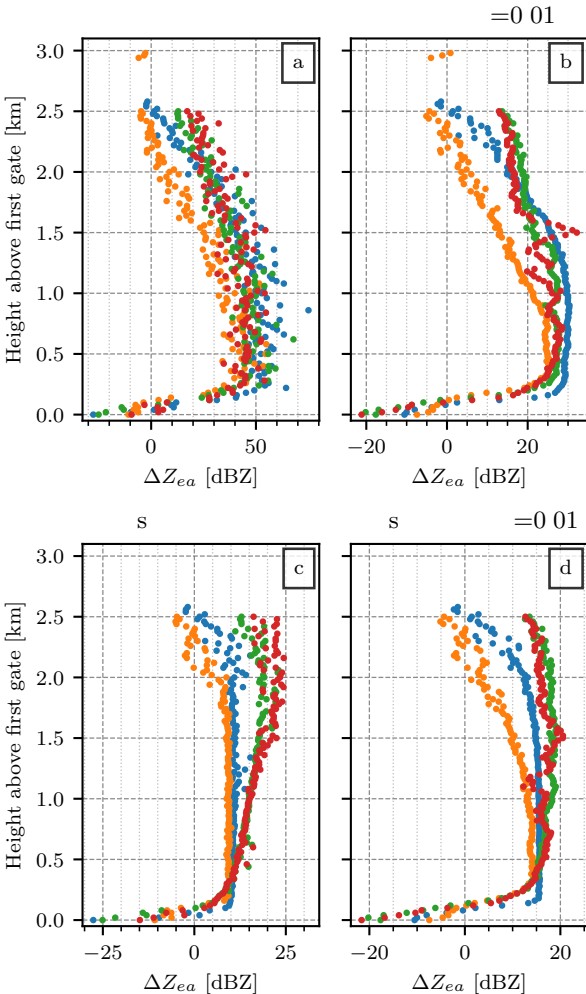

**Figure 6.** Same as the panels c and f of Figure 5, extended to the four datasets and including all products. Colors are assigned to each dataset following the same convention as Figure 3: blue for MRR-PRO 06, orange for for MRR-PRO 22, green for for MRR-PRO 23 and red for ICEGENESIS. Panels a and b display the difference between the profiles of a statistic derived from the original MRR-PRO products and the same statistic computed from the ERUO processing output. Panels c and d show the same quantity, using the postprocessing output instead of the processing one. For panels a and c the statistic used is the minimum value of the $Z_{ea}$ empirical distribution at each range gate, while for panels b and d the statistic is the quantile 0.01. All figures have been truncated at 3.1 km, since the only data points above that height are the ones from the MRR-PRO 06 dataset, visible in Figure 5.



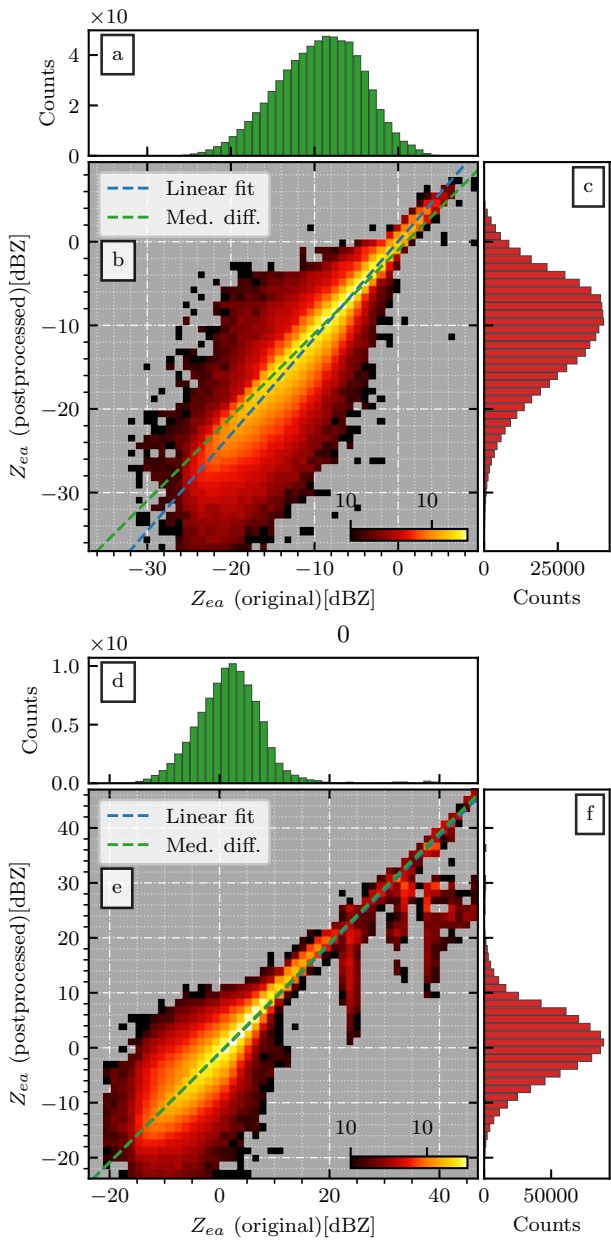

**Figure 7.** Comparison between the $Z_{ea}$ values computed by the original algorithm by Metek (x-axis) and the ones obtained at the end of the ERUO postprocessing (y-axis) for the MRR-PRO 22 dataset (panels a, b, c) and MRR-PRO 06 (panels d, e, f) datasets. The 1-dimensional histogram of the original $Z_{ea}$ value is shown in panel a and d, while the ERUO one is in panel c and f. The color of each bin in panels b and e is proportional to the number of counts in it, with the lowest values being depicted in dark red, and the highest ones in bright yellow.



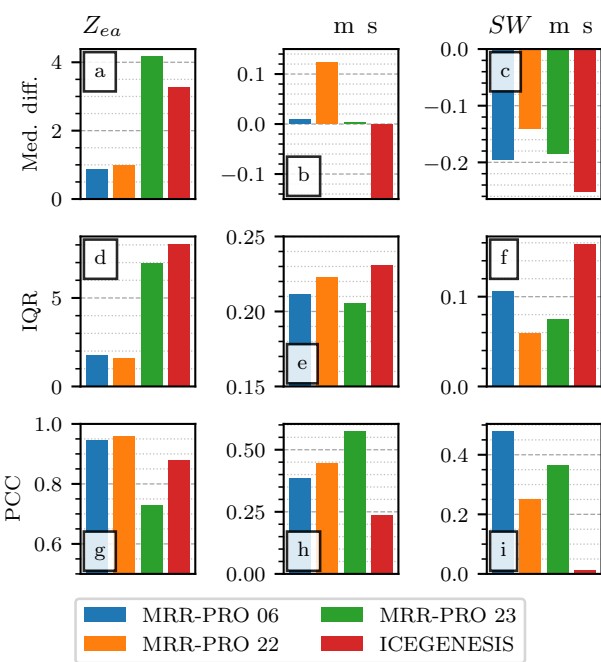

**Figure 8.** Comparison of the ERUO postprocessed products with the variables available in the original MRR-PRO files. The first row of panels (a, b, c) displays the median of the difference between corresponding entries in the two datasets (i.e. original minus ERUO). The second row (d, e, f) shows the interquartile range of the same difference. The third row (g, h, i) displays the Pearson's correlation coefficient, with all values having an associated p-value lower than $4 \times 10^{-22}$. Each column is associated with a different variable, indicated in the title at the top, together with its unit of measurement. Colors are assigned to each dataset following the same convention as Figure 6. The range of the y-axis in panel b has been artificially reduced to allow the visualization of the three datasets collected at PEA. The fourth bar, associated to the ICEGENESIS dataset, reaches the value -18.1 m/s.

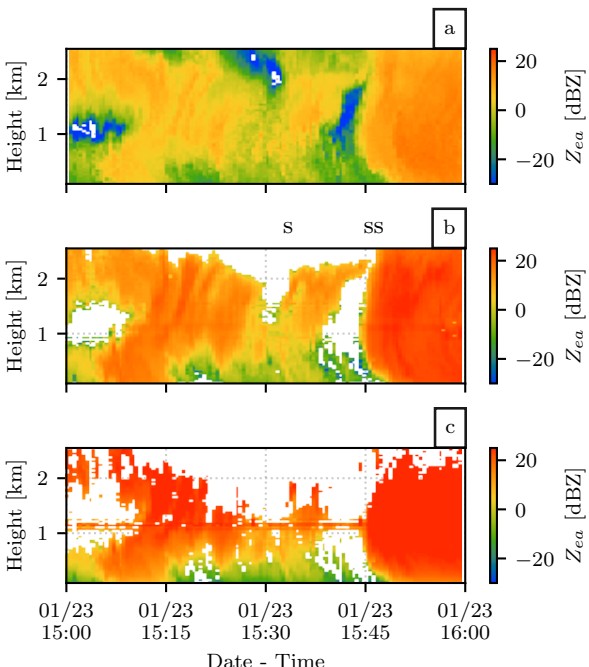

**Figure 9.** Time series of attenuated equivalent reflectivity recorded by WProf (panel a) and the MRR-PRO (original Metek products in panel b, ERUO postprocessing ones in panel c) covering one hour of snowfall during the event on the 23/01/2021.

**Figure 10.** Comparison between the $Z_{ea}$ values recorded by WProf (x-axis) and the MRR-PRO (y-axis) during the ICEGENESIS campaign. The MRR-PRO reflectivity values in panels a, b, and c are the computed by the original algorithm by Metek, while the ones in panels d, e, f are the output of the ERUO postprocessing. The 1-dimensional distribution for WProf, the original MRR-PRO attenuated reflectivity factor and the one derived by ERUO are displayed in panel a (and panel d), panel c and panel f respectively. Density in panels b and e is displayed similarly to Figure 7



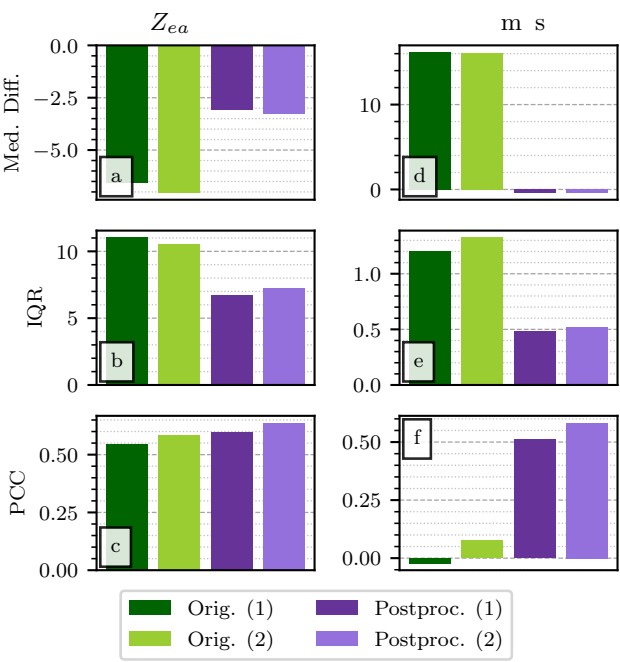

**Figure 11.** Comparison between the ERUO postprocessed products (Postproc.), the variables available in the original MRR-PRO files (Orig.) and the ones recorded by WProf during the ICEGENESIS campaign. The Figure follows the same layout as Figure 8, with the exception of the $SW$ column, which has not been included in the comparison. The numbers (1) and (2) appended to the name of the columns in the legend refers to which radar volumes have been included in the comparison. While (1) refers to the more relaxed condition involving only the dominant hydrometeor type, (2) includes also the threshold (20%) on the proportion of aggregates present in the volume.



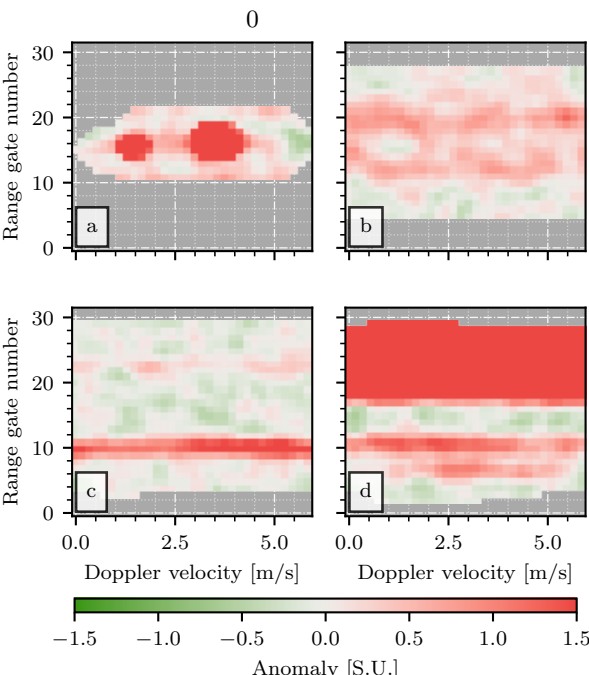

**Figure 12.** Examples of anomalies associated to interference lines, extracted from clear-sky data from the four dataset. The name of the dataset from which each panel has been derived is displayed in its title. The ones displayed are just four example out of the 180 matrices used in the spectrum reconstruction verification.



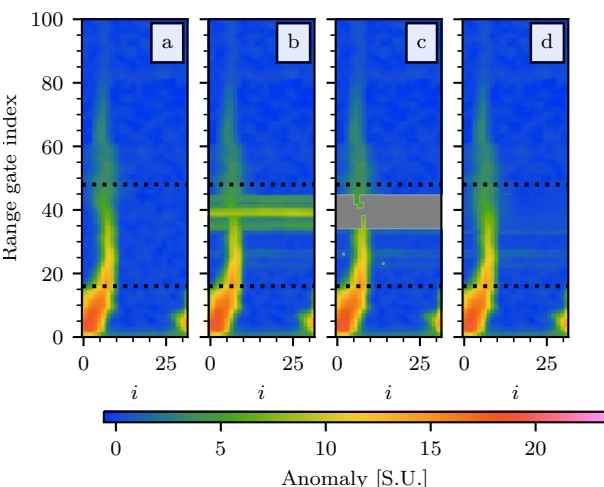

**Figure 13.** Example of reconstruction of the raw spectrum, performed during the verification phase. Panel a shows an example of unaltered spectra collected by the MRR-PRO 06. Once the interference matrix has been applied, the modified spectra are displayed in panel b. Panel c illustrates the identification by ERUO of the region of the spectrum that needs to be reconstructed. Finally, the reconstructed spectra are shown in panel d. The vertical extent of all panels has been truncated at $n = 100$. The horizontal dotted line delimit the region in which the interference line has been added, and over which the skills will be computed.



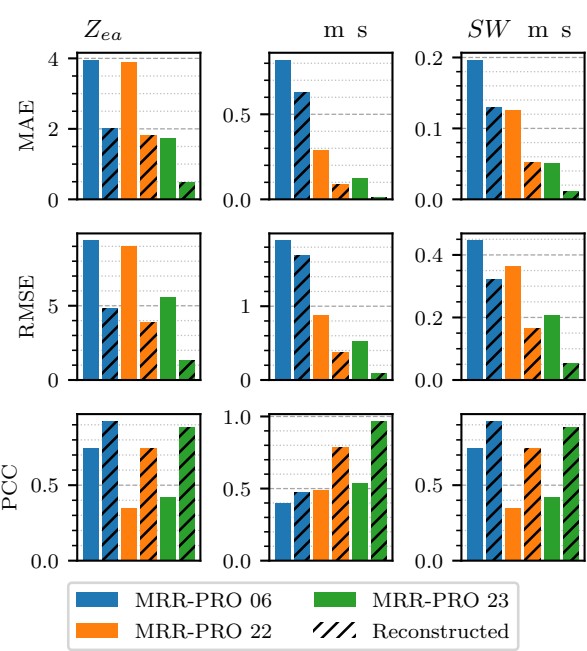

**Figure 14.** Comparison of the variables computed starting from the modified spectra (plain columns) and the reconstructed one (black diagonal hatching) with the ones derived from the original, unaltered spectra. Colors are assigned to each dataset following the same convention as Figure 6. The Figure follows a similar layout to Figure 8, with some difference in the scores: the mean absolute error (MEA) is displayed in the first row, while the root mean square error (RMSE) is shown in the second one.

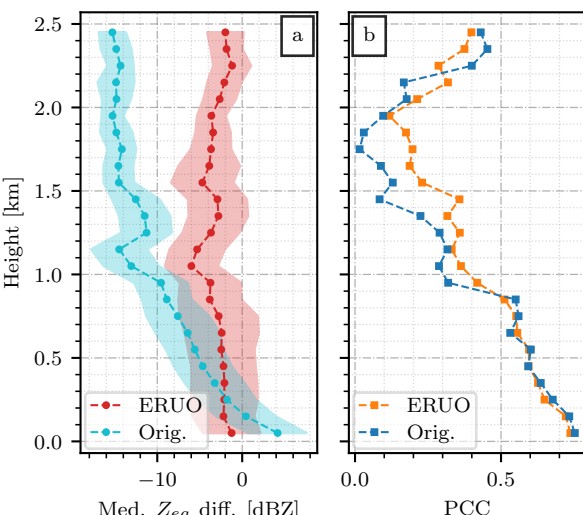

**Figure A1.** Comparison of the the original MRR-PRO $Z_{ea}$ measurements (blue and cyan) and the ERUO postprocessed ones (red and orange) with the $Z_{ea}$ recorded by WProf, repeated for each 100 m section of the profile independently. Panel a displays the change in median difference with height (dots and dashed line), with the IQR represented by a shaded area around the curve. Panel b shows the Pearson correlation coefficient.