# Peer review of "ERUO: a spectral processing routine for the MRR-PRO"

_Atmospheric Measurement Techniques, 2021_

## Referee Comment (RC1)

**Review of ERUO: a spectral processing routine for the MRR-PRO**

by Alfonso Ferrone, Anne-Claire Marie Billault-Roux, and Alexis Berne

November 23, 2021

**1 Short description**

In this paper, the authors introduce an alternative spectral processing system (ERUO) for processing the raw data produced by the new Micro Rain Radar (MRR-Pro). Using ERUO, the authors aim to minimize the effect of interference lines and other issues that affect the MRR-Pro. The comparison between the output from the original software and ERUO shows that ERUO reduced the interference line effects and improved the sensitivity.

The processing system (ERUO) is a good contribution to the radar community, especially for those who work with the MMR-Pro and experienced similar problems. I, therefore, recommend this paper for publication at AMT, but I ask the authors to address the minor issues listed below.

**2 Python package comments**

ERUO is the core of this publication, and it is intended to be publicly available. In order to make it accessible to other users, its documentation needs improvements. In addition to the python scripts (available at: https://github.com/alfonso-ferrone/ERUO), the authors should provide a sample of test files alongside a tutorial where any potential user could be able to test ERUO.

**3** Text comments**

1. section 2.2, line 112:

"... a vertically-pointing W-band (94 GHz) Doppler cloud radar, thereafter referred to as WProf, was also deployed at the airport of La Chaux-de-Fonds, a few meters away from the MRR-PRO. ..."

How far are a few meters away? Was it less than 10, 20 or 50 meters?

2. section 2.3, line 122:

" ... The last radar playing a role in this study is an X-band scanning Doppler dual polarization weather radar (MXPol)  $\dots$  "

What is the frequency of the X-band radar?

3. section 3.1, line 169:"Two examples of the main category are visible in Figure 2 ..."

What are the panels in Figure 2?

4. section 3.1.1, line 182: "Examples of both quantities for the MRR-PRO 06 dataset are shown in Figure 2."

What are the panels in Figure 2?

5. section 3.1.1, line 189:

" More precisely, the beginning of this second part of the profile is moved to the first n in which  $\nabla_n \tilde{\mathbf{S}}(n)$  reaches the median value of all negative  $\nabla_n \tilde{\mathbf{S}}(n)$ ."

Why did you use the median value of all negative  $\nabla_n \tilde{\mathbf{S}}(n)$  to identify  $n_{up}$ ? What would happen if a fixed number of range gates above the point where  $\nabla_n \tilde{\mathbf{S}}(n)$  becomes negative was used to define  $n_{up}$ ? (for example, the first range gate where  $\nabla_n \tilde{\mathbf{S}}(n)$ becomes negative)

6. section 3.1.1, line 191:

"... how the gradient does not reach its typical negative value ..."

What do you mean by typical negative value?

7. section 3.1.1, line 198:

" ... constant is set by default to 3, a value that gives satisfactory results for our datasets. "

What are satisfactory results? How much of the spurious peaks are removed (30%, 50% or 90%)?

8. section 4.1.1, line 408:

"The effects of interference lines on a dataset is clearly visible in both panels,..."

What are the panels?

9. section 4.2, line 656:

"Panel b shows that the ERUO products have a significantly smaller median difference from  $\mathbf{V}^{W}(t, n)$ ."

Panel b from figure 11 does not show the median difference from  $\mathbf{V}^{W}(t, n)$ . It shows the IQR calculated from Ze. Should it be panel d instead?

**4 Figure issues**

1. Figure 2 Panel e :

The horizontal soft red lines are really difficult to see in the PDF and in the printed version. This figure needs an improvement of the contrast.

- 2. Figure 5 Panels b and e: What are S and SS on top of those panels?
- 3. Figure 6:

On top of panels b and d, What is = 0.01? Something is missing On top of panels c and d, what is the meaning of S?

4. Figure 8:

First column: what are the units of  $Z_{ea}$ ? Second column: What is the radar variable? Check the units Third column: Check the units

- 5. Figure 9: What are S and SS on top of panel b?
- 6. Figure 10: What are S and SS on top of panel d?
- 7. Figure 11:

First column: what are the units of  $Z_{ea}$ ? Second column: What is the radar variable? Check the units

8. Figure 12:

The caption says, "The name of the dataset from which each panel has been derived is displayed in its title..." The name of the dataset is not indicated (See the titles).

9. Figure 14:

First column: what are the units of  $Z_{ea}$ ? Second column: What is the radar variable? Check the units Third column: Check the units

**5 Typos**

- 1. Section 3.1.1, line 191: "of  $n_{up}$  for for (repeated for) the MRR-PRO 06"
- 2. Section 4.1.2, line 571: " coefficient in panels g,h (missing space) and i."
- 3. section 4.2, line 645: "attenuation at W-band compared to k-band (K for consistency)."

---

## Author Response (AR1)

**Reply to referee comments**

We thank the reviewers for the valuable comments and suggestions which helped improve the quality of the manuscript as presented in the revised version.

In the following, the comments from the reviewers are written in italic and highlighted in green.
A version of the manuscript with track changes, thereafter referred as latexdiff file, is provided together with the reply. In this file, the part of the text that have been removed are highlighted in red, while the ones that have been added are highlighted in blue.

**1. Overview**

Thanks to the comments and suggestions from the reviewers, the ERUO algorithm underwent some significant modifications. We decided to list them here, at the beginning of the reply, so they are more easily spotted and not lost among the details of the answers to each comment.

These modifications are:

1. The "triplication" of the spectrum, performed during the processing for the identification of the main peaks and to address the aliasing, has been changed. In the current version, at each range gate the spectra from the gate above and below are used, instead of repeating three times the spectrum at the current gate.
2. The standard deviation of the noise is not added anymore to the noise level before subtracting it from the spectrum (during the signal identification). However, this step alone would increase significantly the leftover noise in the final products. Therefore, the following modifications were introduced in the signal estimation:
   ○ signals that are below "noise level + 3*standard deviation" are still excluded,
   ○ isolated signals (width of 1 range gate or 1 velocity bins) are removed before proceeding with the processing.
   The final products appear, from a visual inspection, to have significantly less leftover noise after the processing. This can also be seen in the sensitivity plots in the manuscript, and in the reduced difference between processed and post-processed products.
3. The transfer function reconstruction is now avoided in favor of using an external transfer function, provided by Metek. The reconstruction is still left in the algorithm (and turned off by default), to be used in case the transfer function available in the MRR-PRO file is corrupted and it is impossible to recover the true one by contacting Metek.

**2. Reply to RC 1**

**2.2 Python package comments**

*ERUO is the core of this publication, and it is intended to be publicly available. In order to make it accessible to other users, its documentation needs improvements. In addition to the python scripts (available at: https://github.com/alfonso-ferrone/ERUO), the authors should provide a sample of test files alongside a tutorial where any potential user could be able to test ERUO.*

As suggested by the reviewer, the documentation has been expanded:

• a "User manual" has been added to the repository,

• three example files have been uploaded,

• a video-tutorial that illustrates how to process the example files has been linked in both the "Readme" file and in the manual.

**2.3 Text comments**

1. *section 2.3, line 122:*
   *"...The last radar playing a role in this study is an X-band scanning Doppler dual-polarization weather radar (MXPol) ... "*
   *How far are a few meters away? Was it less than 10, 20 or 50 meters?*
   The distance (approximately 10 meters) has been specified (line 116 of the latexdiff file).

2. *section 3.1, line 169:*
   *"Two examples of the main category are visible in Figure 2 …"*
   *What is the frequency of the X-band radar?*
   The frequency (9.41 GHz)  has been specified (line 126 of the latexdiff file)

3. *section 3.1, line 169:*
   *"Two examples of the main category are visible in Figure 2 …"*
   *What are the panels in Figure 2?*
   The whole line has been re-phrased, and reference to the figure have been removed. The figure was moved in the Appendix A, alongside the discussion of the two preprocessing steps.

4. *section 3.1.1, line 182:*
   *"Examples of both quantities for the MRR-PRO 06 dataset are shown in Figure 2. "*
   *What are the panels in Figure 2?*
   The whole line has been removed, since the discussion of the two preprocessing steps has been moved to the Appendix A.

5. *section 3.1.1, line 189:*
   *" More precisely, the beginning of this second part of the profile is moved to the first n in which $\nabla_n \tilde{S}(n)$ reaches the median value of all negative $\nabla_n \tilde{S}(n)$. "*
   *Why did you use the median value of all negative $\nabla_n \tilde{S}(n)$ to identify $n_{up}$? What would happen if a fixed number of range gates above the point where $\nabla_n \tilde{S}(n)$ becomes negative was used to define $n_{up}$? (for example, the first range gate where $\nabla_n \tilde{S}(n)$ becomes negative)*
   Using the first gate after the gradient becomes negative was the first approach we attempted. Unfortunately, the gradient "stabilizes" (approximately) only few gates after the transition to negative value, and including these first gates negatively impacts the subsequent fit. If we decided instead to use a fixed number of gates (greater than one) above the transition to negative values for the gradient, the fit may theoretically work as well as it does now. However, in this case we would need to choose the exact number of gates to skip. This choice would be as arbitrary as letting the algorithm choose the exact position of $n_{up}$ based solely on the gradient values, as it is done now. The current approach has also the advantage of being able to adapt to how the gradient of the noise floor varies differently in the different datasets.

6. *section 3.1.1, line 191:*
   *"... how the gradient does not reach its typical negative value …"*
   *What do you mean by typical negative value?*
   The phrase has been modified (line 916 of the latexdiff file) to make it slightly more clear in the manuscript. We meant "the negative value of the gradient typical of the upper part of the profile." This negative value is not exactly the same for each dataset.

7. *section 3.1.1, line 198:*
   *" ... constant is set by default to 3, a value that gives satisfactory results for our datasets. "*
   *What are satisfactory results? How much of the spurious peaks are removed (30%, 50% or 90%)?*
   Our formulation at this point was unclear, therefore we added a short clarification in the manuscript (from line 922 in the latexdiff file). The threshold plays an important role in the definition of what interferences will be covered by the mask. Its value is a result of a trade-off between the masking of "major" lines and the need to avoid masking too many of the "minor" ones. Unfortunately, the distinction between "major" and "minor" interference lines is arbitrary, and different users may have different preferences. The more lines are masked, the less interferences will appear in the final products, but the more the "spectrum reconstruction" will have to guess and "reconstruct" to recover

the meteorological signal. We tried to set a threshold that covers the 1-2 major lines in each dataset consistently, while covering only a few (approximately 5) minor ones.

8. *section 4.1.1, line 408:*
   *"The effects of interference lines on a dataset is clearly visible in both panels,…"*
   *What are the panels?*
   The panels (a, d) has been specified (line 547 of the latexdiff file).

9. *section 4.2, line 656:*
   *"Panel b shows that the ERUO products have a significantly smaller median difference from $V^w(t, n)$."*
   *Panel b from figure 11 does not show the median difference from $V^w(t, n)$. It shows the IQR calculated from $Z_e$. Should it be panel d instead?*
   The reviewer is correct, it is panel d (line 743 of the latexdiff file).

**2.4 Figure issues**

During the upload, something happened to the figures, with some titles and labels disappearing partially. This issue did not appear in the pdf file originally uploaded by us. It appeared only once the manuscript was made available on the website for the discussion. We created again the figures, hoping that this time they will appear correctly. Unfortunately, since the figure appear correctly in both the pdf file and during the creation of the manuscript in Overleaf, we have no way to check if the issue persist before the upload.

1. *Figure 2 Panel e :*
   *The horizontal soft red lines are really difficult to see in the PDF and in the printed version. This figure needs an improvement of the contrast.*
   The figure (Fig. A1 in the revised version) has been modified, reducing the limits of the colorbar to allow a better visibility of the fainter lines in panels e and f.
   Additionally, the marker size and line width in panels b and c have been slightly increased.

2. *Figure 5 Panels b and e:*
   *What are S and SS on top of those panels?*
   The labels of the figure were corrupted during the upload, see the note at the beginning of this section.

3. *Figure 6:*
   *On top of panels b and d, What is = 0 01? Something is missing On top of panels c and d, what is the meaning of S?*
   See the note at the beginning of this section.

4. *Figure 8:*
   *First column: what are the units of Zea? Second column: What is the radar variable?*
   *Check the units Third column: Check the units*
   See the note at the beginning of this section.

5. *Figure 9:*
   *What are S and SS on top of panel b?*
   See the note at the beginning of this section.

6. *Figure 10:*
   *What are S and SS on top of panel b?*
   See the note at the beginning of this section.

7. *Figure 11:*
   *First column: what are the units of Zea? Second column: What is the radar variable?*
   *Check the units*
   See the note at the beginning of this section.

8. *Figure 12:*
   *The caption says, "The name of the dataset from which each panel has been derived is displayed in*

*its title...” The name of the dataset is not indicated (See the titles).*

See the note at the beginning of this section.

9. *Figure 14:*
   *First column: what are the units of Zea? Second column: What is the radar variable?*
   *Check the units Third column: Check the unit*

   See the note at the beginning of this section.

**2.5 Typos**

*1. Section 3.1.1, line 191:*

*”of $n_{up}$ for for (repeated for) the MRR-PRO 06”*

*2. Section 4.1.2, line 571:*

*” coefficient in panels g,h (missing space) and i. ”*

*3. section 4.2, line 645:*

*”attenuation at W-band compared to k-band (K for consistency).”*

The three typos have been corrected.

**3. Reply to RC 2**

**3.1 Reply to major comments**

*There as well I would recall the questions and comments of the reviewer...*

1. *Writing: The writing of the article needs to be improved. It is too long because, firstly, the authors get lost in insignificant details. Occasionally, the article appears as if the authors would have gone through the code and verbalized every line. I recommend that the authors go through the whole paper and try to focus on the really relevant steps. For example, the authors describe in L460 how they treat the beginnings and ends of a data set differently. Since the code is published as well, I wonder whether this is really relevant or rather distracting information? I give some more recommendations in the minor comments below but this list is not exhaustive. Secondly, often the authors wrote very verbose and complicated so that the actual relevant information is hidden. As an example, the caption for Fig. 8 is “The range of the y-axis in panel b has been artificially reduced to allow the visualization of the three datasets collected at PEA. The fourth bar, associated to the ICEGENESIS dataset, reaches the value -18.1 m/s.“ It took a while until I realized that the topic of these sentences is not the y-axis, it simply means “Note that the fourth bar, associated to the ICEGENESIS dataset, reaches a value of -18.1 m/s which is outside of the plot's limits.” Scientific writing does not mean to write sentences as complicated as possible to sound smart – it is actually the opposite because simple sentences tend to be clearer and clarity is one of the most important things in scientific writing. Maybe the coauthors can help making the text more straight forward.*

   The writing of the whole manuscript has been revised. In addition to shortening sections and modifying some of them to make them more easily understandable, two major modifications were done to the structure of the manuscript:

   ○ the details of the two iterations of the preprocessing have been moved to an appendix (from line 888 of the latexdiff file), leaving only a description of the purpose and products of this part of the algorithm in the main text;

   ○ the reconstruction of the transfer function has been merged with the already existing appendix, creating the Appendix B (from line 1003 of the latexdiff file), focused entirely on the transfer function issues.

   These two sections were the ones with the highest amount of details unnecessary to the overall comprehension of the algorithm.

The particular phrase in the caption for Fig. 8 (now Fig 7 in the revised manuscript) mentioned by the reviewer has also been simplified.

2. *Symbols: Apparently, the authors tried to save space by using a lot of symbols which makes it even harder to read. Therefore, I strongly recommend to add a list of symbols in an appendix (and promote it early in the text) and to reintroduce the most import symbols when using referring to them in captions or the summary.*
   A list of symbols has been added in the Appendix C (from line 1027 of the latexdiff file).

3. *Noise: The authors attempt to correct for noise and speculate it is related to snow on the antenna (L530f, L557, L763). I don't understand what the authors mean by noise because observations are always noisy. Do they mean an increased noise floor? Where exactly can this be seen in Fig. 5d? And do the authors have a physical explanation how snow on the dish can cause noise? I would think the thermal emissions of snow at 24 GHz should be negligible or is that wrong?*
   The noise was indeed not enough discussed in the original manuscript. In particular:

   ○ The noise we refer to is not just an increase of noise floor, but rather an increase to the variability in the fluctuations of the noise floor for some time steps of the dataset. These fluctuations can be seen as "peaks" in random position of the spectra, appearing at random combination of range gates and velocity bins. In the products available in the original files (the ones computed by the Metek algorithm), these peaks in the spectrum appear as spurious signals the processed variables (Z, VEL, SW, etc). When looking at them in a time-height plot, depending on the dataset considered and on the date, it is possible to notice some "random pixels" appearing:

     ▪ In the lowest range gates for MRR-PRO 06

     ▪ At any height for MRR-PRO 22

     ▪ Similar to MRR-PRO 06, but less often, for MRR-PRO 23 It was this kind of noise in the MRR-PRO 23 data that we observed more often when the antenna was covered in snow.

     In the ERUO products, the increased sensitivity may result in these random peaks appearing more often, since these fluctuations may be comparable in intensity with the weakest detectable meteorological signals. The phenomenon is similar to what happens with the faintest interference lines, which are only sporadically detected in the Metek product, while they would be more continuous in the ERUO products if the spectrum reconstruction is not activated.

   ○ Fig. 5.d (Fig 4.d in the revised manuscript) shows a continuous ridge in the 2-d histogram for the Metek products above the 2 km height. This ridge is close to the minimum sensitivity of the radar, and valid measurements at those heights are not present in the MRR-PRO at different sites. When inspecting time series of MRR-PRO 22 measurements, we can see random pixels with non-NaN measurements in the Zea field available in the original files. These peaks appear at any height (as mentioned in the point above). At lower height their influence in the 2-d histogram is limited by the higher abundance of valid measurements. Above 2 km, instead, these peaks are, together with interference lines, the only available non-NaN measurements, and therefore they dominate the 2-d histogram.
     Since in the original manuscript the explanation was unclear, we re-wrote it (from line 555 in the latexdiff file).

   ○ The speculation on the effect of snow on the antenna on the noise is purely based on observations in the ICEGENESIS campaign. We observed that noise in the low level appeared more often when the antenna was covered in snow. After a manual cleaning of the antenna, noise was still present, but it covered less "pixels" of the time-height plots. We do not have a physical explanation for this phenomenon.
     Any mention of the link between increased noise and snow on the antenna has been removed from the manuscript.

4. *General: I would recommend to start the article with an example showing how interference and "noise" impact the data quality and explain the algorithm based on the example. This is already partly done in Fig. 9, but I would show it much earlier.*
A picture showing how interference lines and noise affect the data has been added in the article (Fig 1 in the revised manuscript) The figure is now referenced in the introduction (line 46, 49 and 64 of the latexdiff file), when interference is first mentioned, in section 3.3.1, which discusses the interference line removal (line 487 of the latexdiff file), and in section 4.1.1 (line 550 and onward).

5. *Section 3.2.3: FMCW radars like the MRR have the quirk that the data appears in the wrong range gate when aliasing is happening. Therefore, the authors must consider this effect when tripling the spectra to account for aliasing effects. See e.g. discussion in Maahn and Kollias 2012.*
The reviewer is correct, and our algorithm was not performing the dealiasing correctly. Now, the algorithm shifts the spectra by +/- 1 range gate before concatenating the copy of the spectra after/before the central one, when tripling it. The procedure is the same as the one discussed in Maahn and Kollias 2012. The new procedure is described in Section 3.2.2 (from line 373 of the latexdiff file).

6. *L415: I'm not sure what is done here. If the authors propose to remove not only mean noise but mean noise plus 3 times its standard deviation from the measurements this would mean that the actual signal is reduced. For small signal to noise ratios this would lead to a negative bias of the algorithm – consistent with the reduced reflectivities for ERUO in comparison to the MRR algorithm (Fig. 7).*
This section has also been modified. It was originally introduced to compensate for the excessive sensitivity of the algorithm in the lowest range gates, which was detecting faint "noise peaks" as valid meteorological signal. This part of the algorithm has been reworked in the following way:

   ◦ The standard deviation is added to the noise floor only to detect which peaks are effectively signal.

   ◦ However, the noise floor that is subtracted from the spectra is the original one (no standard deviation is added to it), therefore it does not lower anymore the Zea value obtained from the integration.

   ◦ Additionally, this operation can be completely removed by the user by setting the flag NOISE_STD_FACTOR equal to 0 in the "config.ini" file.

   ◦ While changing the code, we realized that an optional step could be added to the process in order to remove noise in the processed spectra. Noise appears as isolated peaks, with no continuity along the range axis, spanning few velocity bins. Therefore, if the flag REMOVE_ISOLATED_PEAK_SPECTRUM is set to 1 in the "config.ini" file, the algorithm removes all the "pixels" in the spectrum that would be eliminated by a binary erosion of the "image". Therefore, isolated peaks of a width of 1 in either the range or velocity dimensions are removed before proceeding with the processing. Note that a binary erosion is not performed, and the rest of the signal in the spectra (the ones wider than 1 in either dimension) is left untouched. This operation is entirely optional and can be turned off by the user.

   In the manuscript, the new procedure is described in Section 3.2.4 (from line 444 of the latexdiff file).

7. *L628: If aggregates cause largest differences between backscattering at 24 and 94 GHz, why do the authors accept up to 20% aggregates?*
Setting the threshold to 0% would have removed the large majority of the measurement, making the later comparison less significant. Our intent was just to ensure that aggregates were not dominating the signal in the radar volumes considered. Unfortunately we could not be completely sure that none of the hydrometeors of each volume was large enough to cause Mie scattering at W-band. This is due to the intrinsic uncertainty of the hydrometeor classification/demixing, and to the variability of sizes in each hydrometeor category. In theory, it is unlikely but not impossible to have a crystal with a

larger diameter than an aggregate in one of the radar volumes, and setting the threshold to 0% for aggregates would reduce the measurement pool for the comparison without providing a 100% certainty on the absence of Mie scattering. Instead, by choosing the hydrometeor type that is less likely to cause Mie scattering (crystals) as the dominant one, and limiting the largest ones (aggregates), we tried to reduce the possible impact of Mie scattering on the comparison at the two frequencies.

8. *L689: The authors interpolated the signal in a region without any significant vertical gradients, so the good agreement is not surprising. How does the method perform in more challenging conditions?*
A new example, with a larger gradient, has been chosen to display the spectrum reconstruction. It should be noted that this particular image is displayed as an example, and the figure displaying the skills of the reconstruction in a quantitative way are already presented in Figure 14 (Fig. 13 in the new version).

9. *Fig. 5: To me, it looks like all problems in the MRR data were above 2,5 km, all data was below. If this is true, why was such an extensive detection and reconstruction method developed? And doesn't this mean that the detection of interference and the evaluation of the reconstruction is not very meaningful because it applies to parts of the spectrum without any signal?*
The lack of a figure in the original manuscript illustrating the effect of interference lines and noise on the original measurements is probably what may lead to think that all problems happen above 2.5 km. We hope that the inclusion of an additional figure (Fig. 1), as suggested by the reviewer, may address this misconception.
In particular:

   ○ While the strongest interference lines of the MRR-PRO 06 and MRR-PRO 22 are in the highest range gates, this is not true for the MRR-PRO 23. An important interference line can already be seen for the ICEGENESIS dataset in Figure 9 (Fig. 8 in the revised version) just at about 1 km of height, and such interference is removed by ERUO in the example displayed in that figure. A couple of interference lines, also at about 1 km of height, can be seen for the dataset collected by the MRR-PRO 23 at PEA.

   ○ Random noise (more common for the MRR-PRO 22) appears at all range gate, not only above 2.5 km.

   ○ Minor interference lines also exist below 2.5 km. While they may not appear often in the original MRR-PRO data, the increase in sensitivity makes them more visible in the ERUO product if the spectrum reconstruction is not enabled.

To address the second question:

   ○ The spectrum reconstruction is evaluated by applying interference extracted from clear-sky measurements over precipitation data collected by the three MRR-PRO.
   The interference are applied at a different height for each MRR-PRO, as detailed in the sub-section "Performance of the Spectrum reconstruction". These heights have been chosen for each MRR-PRO because they are unaffected by interference in that specific MRR-PRO. Depending on the height, in each of the three dataset these artificially-added interference lines intersect the precipitation signal at different stages.
   The skills are computed only over the measurements where the interference intersects precipitation signal. Therefore, the reconstruction is evaluated specifically on part of the spectrum with signal.
   This results in the difference in the performances observed for the three MRR-PRO:

     ▪ worst skills for the MRR-PRO 06, for which the interference lines are applied at lower heights and they intersect wider signals, often with larger vertical gradients,

     ▪ best skills for the MRR-PRO 23, for which the interference lines are applied at higher heights, intersect thinner signals, with smaller gradients.

As a side effect, the performance illustrated in Fig. 14 (Fig. 13 in the revised version) can be considered a "worst-case scenario". In a realistic dataset, interference are not overlapping constantly with the precipitation signal. Performance usually improves when the reconstruction is performed for interferences that do not intersect precipitation signals. Therefore, for a realistic dataset we can expect better perfomances overall than the ones shown in Fig. 14 (Fig. 13 in the revised version).

An estimate of the performances in clear sky could not be obtained, since we lack the reference values for the original files (clear sky implies that Zea, VEL and SW are NaN).

Since in the original manuscript this aspect of the comparison was not clear, we added a clarification in section 4.3 (line 799 of the latexdiff file)

**3.2 Reply to minor comments**

- *L16: Given that the paper is only about a method, I'm not sure why the introduction starts with a paragraph about Antarctica.*
  The introduction has been changed, removing most of the discussion related to Antarctica (from line 14 of the latexdiff file).

- *L108: To my knowledge, the TF compensates for a change in receiver sensitivity related to the changing frequency of FMCW radars.*
  The additional information on the TF has been included in the text (line 111 of the latexdiff file).

- *L262: This paragraph reads more like a manual than a scientific paper. I would recommend to remove the paragraph or move this paragraph to an appendix.*
  The paragraph has been removed. The information is still available in the user manual in the GitHub repository of the library.

- *Fig 5: Due to the TF, the sensitivity of FMCW radars typically does not scale with range squared as a normal pulsed radar does. I would recommend to mention this to avoid confusion.*
  The additional information has been included in the caption of Fig. 5 (Fig.4 in the revised version).

- *L200ff: In theory, the change of sensitivity with height of the MRR-Pro can be calculated from your equation 1 because minimum values of H(sig) should be constant with height. Then, the sensitivity at an arbitrary height level can be used to scale with range squared (or n2 x delta r – see your equation 1) and – this is different to a conventional pulsed radar – TF. If the MRR-pro sensitivity scales as predicted by the theory, the S_fit(n) could be obtained more easily.*
  The reviewer is correct, and $S_{fit}(n)$ could be obtained with the alternative approach proposed. However, selecting only a subset of clear sky data, we observed that the noise floor varies in time. This fluctuation is comparable in magnitude to the prominence of a faint interference line (few tenths of S.U.). Computing $S_{fit}(n)$ in the empiric way described in the manuscript may better capture the effect of these fluctuations on the clear-sky estimate of the noise floor, in our opinion. Moreover, the computationally intensive part of the preprocessing is the one that precedes the estimate of $S_{fit}(n)$. Therefore, changing the way in which the latter is computed would only have marginal benefit on the practical execution of the ERUO algorithm.

- *L272: Just a comment, the authors could try using xarray with the dask library which is particularly designed for handling datasets larger than the machines memory.*
  The use of xarray with the dask library will most probably improve some of the execution times of the algorithm. Currently, xarray is only used by the ERUO library in the creation of the processed (and postprocessed) netCDF files.
  However, this modification would require a significant rewriting of the library, while not benefitting the scientific content of the manuscript. We will surely keep it in mind, and in the future such modification may be implemented, without changing the method described in the manuscript.

- *L303: This appears to be a bug by the MRR's firmware. Please report the firmware version of your MRRs somewhere to make sure the broken version is documented.*
  We contacted Metek and the issue has been discussed with them.
  The library has also been modified so that an external transfer function, provided by Metek, can be

used instead of the one available in the MRR-PRO files. The option to use the old reconstruction has been left in the library (but turned off by default), in case it is not possible to recover the correct one by contacting Metek.
In the revised version of the manuscript, the transfer function reconstruction has been merged with the Appendix A, which became Appendix B in the revised version (from line1003 in the latexdiff file).
Mentions to the transfer function reconstruction have been removed from several parts of the manuscript, and the discussion of the transfer function issue has been shortened (lines 6, 59, 290 and onward, 594, 736, 852, 875, 957, 987 of the latexdiff file).

- *Sec. 3.2.2: I find the term "reconstruction" confusing, isn't this simply an interpolation?*
Indeed the function "astropy.convolution.interpolate_replace_nans" performs an interpolation. However, in the documentation online that displays few examples of its usage ( https://docs.astropy.org/en/stable/convolution/index.html ), the operation is sometimes referred to as "reconstruction", especially when the fraction of the image to be recovered is very large. Therefore, we kept the name of the function in ERUO as "spectrum reconstruction", but we added information in section 3.2.2 (line 328 of the latexdiff file), clarifying that the function performs an interpolation.

- *L313: part should be parts*
"part" replaced with "parts" (line 327 of the latexdiff file)

- *L328: What do the authors mean by "kept"? Kept for the mask or kept for valid data?*
The previous formulation was ambiguous, as pointed out by the reviewer, therefore it was changed in the new version of the manuscript (line 343 of the latexdiff file). We meant "the ones kept in the mask".

- *L329: capital SU*
"s.u." has been changed to "S.U." (line 345 of the latexdiff file)

- *L370: This means that only the moments of the most significant peak are estimated. This is totally fine but should be mentioned.*
While this is true for the specific example shown in Fig. 4.c (Fig, 3.c in the revised version), in general ERUO does not keep only the main peak. When deciding which part of the spectrum is kept unmasked to proceed with the processing (the operation illustrated in Fig. 4.d), eventual secondary peaks (satisfying the conditions listed in the manuscript) are not removed.
If at a specific range two peaks have been recorded, the unmasked section is set as the union of the velocity bin range spanned by the limits of both peaks.
Then:

  ○ If the resulting unmasked region is larger than $m$, velocity bins are masked at the extremes, starting from the ones with the lowest value,

  ○ If the resulting unmasked region is smaller than $m$, velocity bins are unmasked at the extremes starting from the ones with the largest value.

Given the low velocity resolution, the detection of clearly-separated bimodality in the spectrum is extremely rare. The overlap of two peaks is met most often for peaks associated with spurious "noise" fluctuations in the raw spectrum.
Since this aspect of the algorithm was not clear in the original version of the manuscript, we modified part of Section 3.2.2 (line 392 in the latexdiff file) and Section 3.2.3 (from line 413 of the latexdiff file).
On the same topic, while implementing the modifications to the algorithm listed above, we also changed slightly the peak detection. A completely optional limit on the maximum number of peaks detected at each range gate has been added to the algorithm. This limit has been set by default to 6, but it can be changed by the user by modifying the flag MAX_NUM_PEAKS_AT_R in the "config.ini" file.

- *L426: Use equal sign for 0.92 because I assume this exact value is used in the code.*
The sign has been changed to "=" (line 458 of the latexdiff file).

- *Sec. 5: This section can be drastically cut. Focusing on the relevant information here would also allow the reader to identify the main conclusions of the paper.*
  The section has been reduced in length, removing the less useful information (from line 825 of the latexdiff file).

- *General: When referring to figure panels, please repeat the figure number (i.e. "3.a" instead of "a"). This is particularly important when discussing multiple figures at once.*
  The new naming convention for panels suggested by the reviewer is indeed much clearer than the one used in the previous version of the manuscript. We adopted the suggested naming convention (figure_number.panel_letter) in the revised manuscript.

- *All figures: Something went wrong with the figures, letters are missing. E.g., for Fig. 5, the title is: "06 (line break) s          ss"*
  The issue with the titles/labels did not appear in the pdf file originally uploaded by us. It appeared only once the manuscript was made available on the website for the discussion. It appeared only once the manuscript was made available on the website for the discussion. We created again the figures, hoping that this time they will appear correctly. Unfortunately, since the figure appear correctly in both our local copy of the pdf file and during the creation of the manuscript in Overleaf, we have no way to check if the issue persist before the upload.

- *All figures: Add legends of the lines to all figures.*
  Legends have been added to the lines in the figures.

- *Figs. 8 & 11: Are these figures and the related discussions really required in the main text?*
  These figures (Fig. 7 and 10 in the revised version) are the only place in which ERUO is compared quantitatively with the original MRR-PRO products and WProf.
  The first figure gives us an estimate of how much ERUO deviates from the original Metek products. This is valuable information in our opinion, because it shows that the deviation is not very large in absolute terms (0.5 dBZ), and provides to the reader an estimate on what to expect if they apply ERUO to a dataset of MRR-PRO measurements.
  It would be harder to estimate this quantity by eye from the 2-dimensional histogram of Fig 7, since the axes span several tens of dBZ.
  The second figure addresses a possible issue with the spectrum reconstruction and improved sensitivity. In both cases, ERUO recovers some signal that is not in the original Metek product and we cannot test their validity in the comparison of Fig 8 (now Fig. 7). So, Fig. 11 (now Fig. 10) uses WProf as point of comparison, since this radar sees much fainter signals. It shows that, on average, the deviation is large, but not larger that what it was for the original Metek files.
  Both figures in our opinion provide valuable information that cannot easily be seen in the 2-d histograms.

---

## Referee Report (RR1)

**Review of ERUO: a spectral processing routine for the MRR-PRO**

by Alfonso Ferrone, Anne-Claire Marie Billault-Roux, and Alexis Berne

February 8, 2022

**1 Short description**

In this paper, the authors introduce an alternative spectral processing system (ERUO) for processing the raw data produced by the new Micro Rain Radar (MRR-Pro). ERUO aims to minimise interference lines and other issues that affect the MRR-Pro. Comparison between the output from the original software and ERUO shows that ERUO reduced the interference line effects and improved the sensitivity.

The processing system (ERUO) is a good contribution to the radar community, especially for those who work with the MMR-Pro and experienced similar problems. I, therefore, recommend this paper for publication at AMT, but I ask the authors to address the minor technical issues listed below.

**2 Typos**

1. Section 3.2.1, line 197:
   "-$\mathbf{S}^{(cs)}(t_j, i, n)$ The first guess of (missing period)"

2. Appendix B, line 770:
   "a normal transfer function function (repeated function)"

3. Appendix C :
   "$\mathbf{EXC}(t, n)$ Matrix of the **measurments** (typo)"

4. Appendix C :
   "$\mathbf{S}^{(cs)}(t, i, n)$ Clear-sky estimate of the power **retur**(typo)"

---

## Author Response (AR2)

**Reply to referee comments**

We thank the reviewers for the corrections and suggestions which helped to clarify some sections of the manuscript and, in general, improve the quality of the text.

In the following, the comments from the reviewers are written in italic and highlighted in green.
A version of the manuscript with track changes, hereafter referred to as latexdiff file, is provided together with the reply. In this file, the part of the text that have been removed are highlighted in red, while the ones that have been added are highlighted in blue.

**1. Reply to RC 1**

**1.2 Typos**

All typos listed by the reviewer have been corrected. The changes can be observed at the following lines of the diff file: line 208, 812, Appendix C (835).

**2. Reply to RC 2**

**2.1 Technical comments**

Thank you for the information regarding the Doppler velocity range of the MRR-2 and the lack of R^2 dependence being a particularity of the MRR and not of all FMCW radars.

The suggested changes at lines 95 and 140 of the latexdiff file (originally lines 94 and 141) have been implemented.

As noted by the reviewer, Figure 2 was not clearly displaying the transfer function reconstruction as optional. Therefore, we added a decision block before this step, highlighting that the default option is to not execute the reconstruction.

**3. Reply to RC 3**

**3.1 Major comments**

Thank you for all your suggestions, which helped to improve the "Method" section of the manuscript.

*Manuscript is very difficult to read. The problem is not an English language issue, nor a grammar issue.*
Even though the English language and grammar are not pointed out as problematic, the difficulty in reading the manuscript mentioned in the comment prompted us to perform a grammar check of the whole text. Several mistakes scattered in the text have been found and corrected, as visible in the latexdiff file.

*I believe the difficultly is that the manuscript is not written as a technical journal article describing the work that was performed. The manuscript seems to switch between a technical journal article, a software manual, and a personal history. While all are interesting topics, the manuscript should focus on being a technical journal article suitable for AMT. Writing in one style is difficult. But the end result will be a focused manuscript describing the work that was performed. I suggest the authors focus on one MRR data set and rewrite Section 3 with the focus on describing "the work that was done." Note that I find Section 4 easier to read than Section 3. I believe that is because Section 4 describes details of the four data sets, which causes Section 4 to be more focused toward a technical article than Section 3.*
We fully agree on the presence of major issues in Section 3, as pointed out by the reviewer.

Firstly, in the version of the manuscript that we previously submitted, we provided advice to a hypothetical user on how to handle scenarios different from the ones presented in the manuscript.

Such mentions have been removed, as visible in the following lines of the latexdiff file:
- line 150
  The user is advised to display ˜S(i,n), and look for the signature of precipitationin this matrix.

- line 155
  This value can be easily changed by the user in the dedicated section of the configuration file of the ERUO library

- line 203
  It should be noted that this part of the processing is optional, and it can be avoided by setting the flag controlling it in the configuration file equal to 0. However, we strongly recommend running the spectrum reconstruction, to keep the output products as clean as possible

- line 355
  At the end, the quantities $Z^{(Proc)}_{ea}(t,n)$, $V^{(Proc)}(t,n)$, $SW^{(Proc)}(t,n)$, $SNR^{(Proc)}(t,n)$, together with the noise floor and level are saved in a file, at the location specified by the user in the configuration file. The optional "quickplots" routine included in the ERUO library can provide a simple visualization of some of these products.

- line 370
  Therefore, the user can decide to execute both of them, to skip one or to ignore the whole postprocessing.

- line 395
  Therefore, the user may be required to change their value in the configuration file

- line 662
  This behavior can be controlled by the user by setting lower thresholds in the configuration file

Suggestions on how to handle the specific cases are still available in the User manual on GitHub or in the video tutorial.

As suggested by the reviewer, the text required more focus on the work that was performed and on the ERUO method. To achieve this, we decided to remove the vague mentions to what we encountered "in our case". The style of the whole section has been modified to be more impersonal, mentioning specificities of the four MRR-PRO datasets only when strictly necessary. Moreover, all instances of the possibly ambiguous phrasing "in our case" have been removed from the manuscript.

While this type of modifications are placed all thorough section 3, the more evident example can be seen at the following lines of the latexdiff file:
- line 149
  This line is presented by the reviewer as example.
  *For example, the paragraph on lines 145-151 starts with the sentence, "In our case, the signature of precipitation was never visible in this matrix, being instead relegated to lower quantiles." The phrase "In our case" refers to a personal history, not a technical description. The reader needs to know which data set you are talking about when you say, "In our case" (i.e., which case?). and the characteristics of that data set. The phrase "signature of precipitation" is not described. What is a signature of precipitation? I would expect precipitation to produce an increase in power in the radar velocity spectra when compared to clear-sky conditions, yet, the end of the sentence implies that precipitation signal has a decrease in signal power and "relegated to lower quantiles". The phrase "never" is not appropriate for technical journal articles.*
  We removed first two sentences of the paragraph, highlighted in red in the following text.
  In our case, the signature of precipitation was never visible in this matrix, being instead relegated to lower quantiles. However, it may be possible that other datasets contain a higher proportion of precipitation measurements.
  The sentence has been changed to the following text, highlighted in blue.

In all the four datasets presented in section 2, no couple *(i, n)* experiences precipitation for more than 50% of the duration of the campaign. Due to this relative scarcity of precipitation, the usage of the quantile 0.5 (median) to compute ˜$S(i,n)$ is an adequate choice to isolate the non-meteorological background.

In the newer version, the term "in our case" has been substituted by a clearer mention to the four MRR-PRO datasets. The terms "signature of precipitation" and "never" are no longer used.

The reason why the median is used to compute ˜$S(i,n)$ is linked directly with the frequency of precipitation in the four datasets (no couple *(i, n)* experiences precipitation for more than 50% of the duration of the campaign). Additionally, the objective of the ˜$S(i,n)$ matrix is more explicitly stated (isolate the non-meteorological background).

- line 329

  In our case, the MRR-PRO 06 dataset was particularly affected, with spurious signals detected for almost the entirety of the time series collected in clear-sky conditions.

  Similarly to the previous point, this sentence did not provide useful information on "the work that was done". Additionally, it can be seen as redundant, since section 4 discusses the 2-dimensional distribution of the attenuated equivalent reflectivity factor from the MRR-PRO dataset, mentioning the same issue with spurious signal.

**3.2 Specific comments**

2. *Section 3.1 Preprocessing. What is the purpose of the preprocessing? This sections starts with stating the three output products (i.e., IM, BC, and P) that will be produced. Yet, there is not a general description of why the spectra need to be preprocessed. In other words, what is the problem being solved? Stating the purpose at the beginning of the section will help the reader understand the processing steps performed in this section.*

   The section now starts with a sentence that briefly explains the aim of this stage of the algorithm (line 135 of the latexdiff file):

   The preprocessing aims to identify regions in the spectra that are systematically affected by artifacts and to produce a series of products to assist in the processing of the data files.

3. *Line138. Why does the spectrum power decrease at the ends of the spectra? I find this to be an interesting problem. The questions that come to my mind include: What signal processing step causes this decrease? Does this occur for all spectra including with and without precipitation? Is the amount of power decrease on the ends dependent on the total signal power in the spectrum? How does the MRR manufacture handle this issue in their standard processing? These questions should be addressed to determine whether your correction method is appropriate for the issue observed in the raw spectra.*

   The cause of this power drop is now explained, and its appearance in both precipitation and clear-sky data is mentioned (line 141 of the latexdiff file):

   This drop is caused by the filtering performed by the algorithm of Metek, as described for the MRR-2 by Maahn and Kollias (2012). This behavior is visible in both precipitation and clear-sky measurements.

   Answering the more specific questions:

   - *Is the amount of power decrease on the ends dependent on the total signal power in the spectrum?*
     The power drop is independent of the total signal power in the spectrum.

   - *How does the MRR manufacture handle this issue in their standard processing?*
     It is not explicitly mentioned in the manual, therefore we could not include it in the manuscript. However, in the IMProToo algorithm for the MRR-2, a similar issue with the measurements at the border of the spectrum is handled by interpolating the spectra around the v_ny value. This interpolated result was used instead of the original spectrum in the few gates at the beginning and end of the Doppler velocity range.
     While this is a valid solution, in our opinion it may artificially lower the meteorological signal in

cases in which the location of the peak falls in one of the velocity bins affected by the issue.

4. *Line 140. "…a guess of the signal recorded...". Choose a different word than "guess", unless this is a "first guess" of an iterative process.*
The phrasing "a guess of the signal" has been changed to "the typical return".

5. *Line 166. "Examples of interference lines of both types of are displayed in subsection 4.3, alongside a description of how the ERUO library handles their presence and a discussion on some possible impacts on the final radar variables." Please show an example of interference lines in this section and show how the preprocessing steps discussed in this section removes those interference lines.*
We agree on the importance of a visualization of the interference lines in this section, so we included an additional figure, displaying their appearance in two of the datasets.
However, the removal of the interferences is performed by the "spectrum reconstruction", part of the processing (section 3.2.1). The current explanation of the method follows the order in which the components of ERUO are executed, therefore we did not include information from section 3.2.1 in the preprocessing section.

6. *Section 3.2.1, lines 182-231. "The first step of the spectrum-by-spectrum processing can be considered as the most delicate one of the whole ERUO library." Can you show an example spectrum profile showing the masking procedure and the spectrum reconstruction? Also, since this processing is complicated to describe in words, can a flow diagram be used to describe the logic?*
A visual representation of the reconstruction is illustrated in section 4.3 (verification). To avoid repetition, we now simply included a reference to section 4.3 in the main text.
Unfortunately, we could not find a clear way to summarize the procedure in a flow diagram. We decided, instead, to change the phrasing of some part of the sections, in an attempt to make it more understandable.

7. *Line 221: What is the black box named "astropy.convolution.interpolate_replace_nans" function actually doing? Please describe the analysis or method applied to the data set, not the tool that was used with the "default" values. You can reference this tool and include it in the reference section. But the text should describe the method applied to the data set*
The sentence has been rephrased, mentioning the kernel interpolation before referencing the Astropy function.
The reason why the parameters have been chosen is now mentioned more clearly (from line 237 of the latexdiff onward):
The kernel type used in the procedure is a Gaussian one. Its combination of standard deviation and kernel size has been chosen for its ability to capture the typical shape of the meteorological signal observed in the four datasets.
In particular, the kernel standard deviation has been fixed to 1 along the $i$-axis. A higher value causes an artificial broadening of the reconstructed peaks compared to the precipitation signal directly above and below. On the $n$-axis, the kernel needs to be large enough to contain non-NaN values at both its extremes to perform a meaningful interpolation. To satisfy this condition, the standard deviation size along this axis is set equal to the number of consecutive range gates containing at least one NaN divided by a scaling factor, set by default to 3.

8. *Lines 212-231. I find this section hard to read because there are a lot of descriptions of NaNs and threshold values. Can this section be shortened?*
We agree with the statement: the section is indeed hard to read and full of technical details. However, by removing part of this section, a hypothetical reader willing to replicate the procedure for another radar would not have access to all the information needed to implement the procedure, reducing slightly the usefulness of the whole explanation.
However, as mentioned in the answer to comment 6, the text section 3.2.1 has been modified in an attempt to make it more clear and understandable.

9. *Line 233. "…and converted to linear units." I am confused. Is this saying that in the previous sections, the spectra were in logarithmic units? I thought the spectra in those sections were processed in linear Spectra Units (S.U.) as described in lines 91, 198, 206, and 216. So, is the work before line 233 performed in linear or logarithmic units? I am confused.*
We apologize for the confusion. The preprocessing is performed in logarithmic units. The first

appearance of "S.U." in the manuscript is now labeled as "logarithmic spectral units", while the first appearance of "s.u." (in the section "Peak detection and dealiasing") is labeled as "linear spectral units".

10. *Line 649. "…variables Zea, V, SW, and SNR." Expand the names of the variables in the conclusion so that the reader does not need to search the manuscript for their definition.*
    The acronym of each variable is now preceded by the full name of the variable (line 686 of the latexdiff file).